# Fate and propagation of endogenously formed Tau aggregates in neuronal cells

Patricia Chastagner[1,†], Frida Loria[1,†,‡‡], Jessica Y Vargas[1], Josh Tois[1], Marc I Diamond[2], George Okafo[3], Christel Brou[1,‡] & Chiara Zurzolo[1,*,‡] (iD)

## Abstract

Tau accumulation in the form of neurofibrillary tangles in the brain is a hallmark of tauopathies such as Alzheimer's disease (AD). Tau aggregates accumulate in brain regions in a defined spatiotemporal pattern and may induce the aggregation of native Tau in a prion-like manner. However, the underlying mechanisms of cell-to-cell spreading of Tau pathology are unknown and could involve encapsulation within exosomes, trans-synaptic passage, and tunneling nanotubes (TNTs). We have established a neuronal cell model to monitor both internalization of externally added fibrils, synthetic (K18) or Tau from AD brain extracts, and real-time conversion of microtubule-binding domain of Tau fused to a fluorescent marker into aggregates. We found that these endogenously formed deposits colabel with ubiquitin and p62 but are not recruited to macroautophagosomes, eventually escaping clearance. Furthermore, endogenous K18-seeded Tau aggregates spread to neighboring cells where they seed new deposits. Transfer of Tau aggregates depends on direct cell contact, and they are found inside TNTs connecting neuronal cells. We further demonstrate that contact-dependent transfer occurs in primary neurons and between neurons and astrocytes in organotypic cultures.

**Keywords** autophagy; Intercellular spreading; prion-like seeding; Tau aggregates; tunneling nanotubes

**Subject Category** Neuroscience

## Introduction

The progressive accumulation of aggregated misfolded proteins is a common phenotype observed in several neurodegenerative disorders. In Alzheimer's disease (AD), the hallmark proteins are extracellular amyloid-beta deposits (senile plaques) and intracellular inclusions (neurofibrillary tangles), which consist of microtubule-associated protein Tau, hereafter named Tau protein. These ordered assemblies have properties similar to amyloid fibrils and may propagate throughout the brain in a prion-like manner. Like prions, Tau fibrils act as templates for conversion of native protein to a fibrillar form, initiating a self-amplifying cascade, and can spread from their initial production site to other areas in the brain, following well-defined pathways (Jucker & Walker, 2013). In AD brains, phosphorylated Tau accumulates first at the noradrenergic locus coeruleus (Braak *et al*, 2011; Grinberg & Heinsen, 2017), with seeding first detected at entorhinal/limbic areas (Kaufman *et al*, 2018), before it spreads in a stereotypical manner to interconnected neocortical regions (Jucker & Walker, 2011). Tau accumulation is detected in the brain at least one decade before the appearance of the clinical symptoms of AD, by which time the proteins have spread progressively throughout patients' brains (Holtzman *et al*, 2011). The degree of tauopathy in the brain correlates with the cognitive decline in AD (Braak & Braak, 1991), suggesting that spreading of Tau deposits could be associated with disease progression. More recently, it has been shown that the prion-like activity of insoluble Tau, rather than its bulk accumulation, was inversely correlated with longevity of patients, highlighting the importance of understanding how Tau aggregates are generated and spread (Aoyagi *et al*, 2019). Moreover, dominantly inherited mutations in the *microtubule-associated protein Tau* (*MAPT*) gene, which encodes Tau protein, cause frontotemporal dementia and parkinsonism linked to chromosome 17 (FTDP-17T). Dysfunction of Tau is also involved in multiple disorders linked to neurodegeneration and dementia, collectively termed tauopathies (Goedert & Spillantini, 2017).

Tunneling nanotubes (TNTs) represent a newly discovered mechanism of cell-to-cell spreading of different cargoes, including abnormal proteins. TNTs were first observed in 2004 (Rustom *et al*, 2004) and are comprised of F-actin-containing channels that connect cells over large distances. They represent a novel mechanism for long-range intercellular communication operating in different cell types and in many diseases (Abounit & Zurzolo, 2012;

1   Unité de Trafic Membranaire et Pathogenèse, Institut Pasteur, Paris, France
2   Center for Alzheimer's and Neurodegenerative Diseases, Peter O'Donnell Jr. Brain Institute, University of Texas Southwestern Medical Center, Dallas, TX, USA
3   GlaxoSmithKline, Stevenage, UK
    *Corresponding author. Tel: +33 0145688277; E-mail: chiara.zurzolo@pasteur.fr
    †These authors contributed equally to this work
    ‡These authors contributed equally to this work
    ‡‡Current address:   Laboratorio de Apoyo a la Investigación, Hospital Universitario Fundación Alcorcón, Madrid, Spain

Baker, 2017). Unlike other filamentous bridges (e.g., filopodia, cytonemes), TNTs directly connect the cytoplasm of distant cells (Sartori-Rupp et al, 2019) and selectively transfer a wide variety of cellular materials, e.g., cytoplasmic molecules, miRNA, vesicles, and organelles. In addition, TNTs can be hijacked by various pathogens, such as bacteria, viruses, and protein aggregates

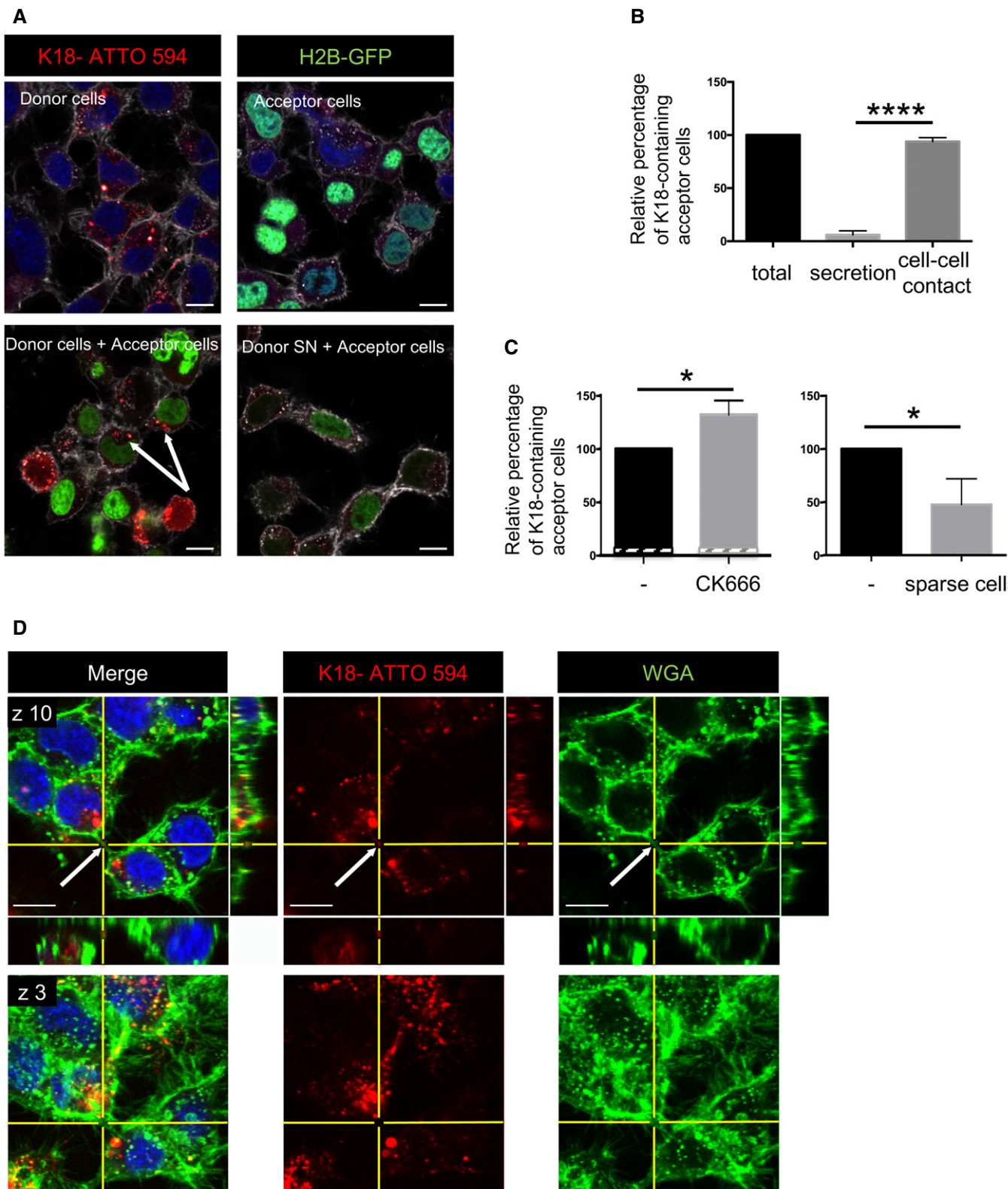

**Figure 1.**

**Figure 1. Spreading of K18-ATTO 594 fibrils in CAD cells.**

A   Transfer of K18-ATTO 594 fibrils from donor cells to H2B-GFP-expressing acceptor cells. Representative confocal images of each population are in the upper panels, and below are pictures after 24 h of coculture of the two populations (left) and of acceptor cells treated with conditioned medium from donor cells for 24 h (SN, right). In white is the cell membrane labeling with WGA (wheat germ agglutinin) coupled to Alexa 647. The arrows point to acceptor cells containing fibrils, scale bars are 10 μm.

B   Quantification by flow cytometry of the percentage of K18-ATTO 594-positive acceptor cells after coculturing donor and acceptor cells (total), or culturing acceptor cells with donor-conditioned medium for 24 h (secretion). The total transfer is arbitrarily set at 100%, and cell-to-cell contact transfer is calculated by subtracting secretion transfer from total transfer. Data represent the means (+ SD) of four independent experiments, with statistical analysis by two-tailed unpaired *t*-test (****$P = 4.64E-08$).

C   Quantification by flow cytometry of the relative percentage of K18-ATTO 594-positive acceptor cells upon treatment with CK666 during the coculture. Left, data represent the means (132%) + SD, normalized to non-treated coculture arbitrarily set at 100%, of three independent experiments, with statistical analysis by two-tailed unpaired *t*-test (*$P = 0.015$). The hatched area of each bar represents the part due to secretion (respectively, 4.8 and 4.3% in the absence and presence of CK666). Right is the same analysis when the cells were cultured in sparse conditions, not allowing cell-to-cell contacts (mean = 47.7%, three independent experiments, *$P = 0.020$).

D   Representative confocal images (40×) of CAD cells treated with 1 μM K18-ATTO 594 fibrils, 24 h after fibril addition, and fixed and stained with WGA-Alexa 488 (green) and DAPI (blue in the merge panels). Bottom panels are a bottom slice corresponding to the substrate-attached surface of cells (z3), upper panels correspond to slice 10 of the same picture (z10), not attached to the substrate. On the right and below z10 pictures are the orthogonal views (*xz* and *yz*) of the same region covering 27 slices over 11 μm in total. The arrows point to red fibrils into a WGA-positive TNT. Scale bars are 10 μm.

(Gousset & Zurzolo, 2009; Abounit *et al*, 2016a; Victoria & Zurzolo, 2017; Ariazi *et al*, 2017).

Previous research has suggested that exogenously added Tau fibrils in neuronal cocultures can propagate through TNTs (Tardivel *et al*, 2016; Abounit *et al*, 2016b). However, data supporting the dynamics of spreading and subsequent seeding are still missing, particularly as the protein aggregates used in those studies were not formed endogenously. Here, using neuronal cell lines, as well as primary neurons and organotypic cultures, we analyzed the subcellular compartment where fibrils could be processed and eventually propagated. Then, we created a neuronal biosensor cell model (Holmes *et al*, 2014) expressing the microtubule-binding domain of Tau fused to a fluorescent marker (RD-YFP) to extract quantitative parameters regarding time and efficiency of new aggregates seeding, induced by either synthetic or natural AD-derived Tau fibrils. We demonstrate that both synthetic and natural fibrils can seed the formation of endogenous Tau aggregates. The latter escape degradation by both autophagy and proteasome. Finally, we show that endogenously formed Tau aggregates, composed of RD-YFP or of full-length Tau, and appearing when cells were challenged by synthetic fibrils, use a cell contact-dependent manner, possibly TNTs, to propagate in different neuronal culture models.

## Results

### Tau fibril transfer between neuronal cells via cell-to-cell contact

To study how Tau fibrils propagate between cells, we generated fluorescently labeled Tau fibrils from monomeric K18 proteins (containing the aa 244–372 of human Tau protein, also called RD domain), shown to have similar structural and physicochemical properties regarding fibril formation compared with full-length Tau (Ait-Bouziad *et al*, 2017). Purified K18 monomers were assembled into insoluble aggregates in the presence of heparin, purified by ultracentrifugation, and labeled with ATTO 594. The quality of the assembled fibrils was checked by SDS–PAGE and Coomassie staining, as well as by thioflavin T assay as shown before (Fig EV1A) (Li & Lee, 2006).

Mouse neuron-like CAD cells challenged with the fibrils by cationic lipid-mediated transfection (Lipofectamine) exhibited red puncta inside the cells. These puncta were resistant to trypsin treatment, showing that the fibrils were internalized by approximately 50% of the cells and were not stuck on the cell surface (Fig 1A (top left) and Fig EV1B). Since exogenously added fibrils from full-length Tau have been previously shown to transfer between cells in various cell lines including CAD cells, and were observed inside TNTs (Abounit *et al*, 2016b), we assessed whether our K18-ATTO 594 fibrils could be transferred between CAD cells in a classical coculture assay. Here, CAD donor cells, first challenged overnight with K18-ATTO 594 fibrils, were trypsinized and replated in coculture with a population of acceptor cells expressing H2B-GFP (Fig 1A top right and bottom left panels). This protocol assured that there were no residual fibrils stuck on the cell surface of donor cells and no residual Lipofectamine 2000 (Fig EV1B). After 18 h of coculture, we performed immunofluorescence (Fig 1A) and flow cytometric analysis (Fig 1B) to quantify the H2B-GFP-expressing acceptor cells positive for K18-ATTO 594 (Abounit *et al*, 2015). Acceptor cells (cultured in parallel) were also challenged for 18 h with conditioned media from donor cells (named donor SN, for supernatant in Fig 1A bottom right panel), to directly monitor the amount of K18-ATTO 594 fibrils transferred via a secretory mechanism only (i.e., not dependent on cell-to-cell contact; named secretion in Fig 1B). Figure 1A is a representative immunofluorescence, where after coculture (which allowed cell-to-cell contact), K18-ATTO 594-positive spots were visualized in acceptor cells (indicated by arrows in the bottom left panel). In contrast, very few red puncta were detected when acceptor cells were challenged with the supernatant from donor cells (bottom right panel). Quantitation of the data was performed by flow cytometric analysis, which showed that cell-to-cell contact-dependent transfer accounted for 94% of the total transfer, while the percentage of transfer through supernatant was negligible (Fig 1B). To assess whether this cell contact-dependent transfer could be correlated with a TNT-dependent mechanism, we analyzed whether compounds or culture conditions that modulate TNT formation affected Tau fibril transfer. When the cells in coculture were treated with CK666 (an Arp2/3 inhibitor known to decrease filopodia formation and to increase TNT formation (Delage

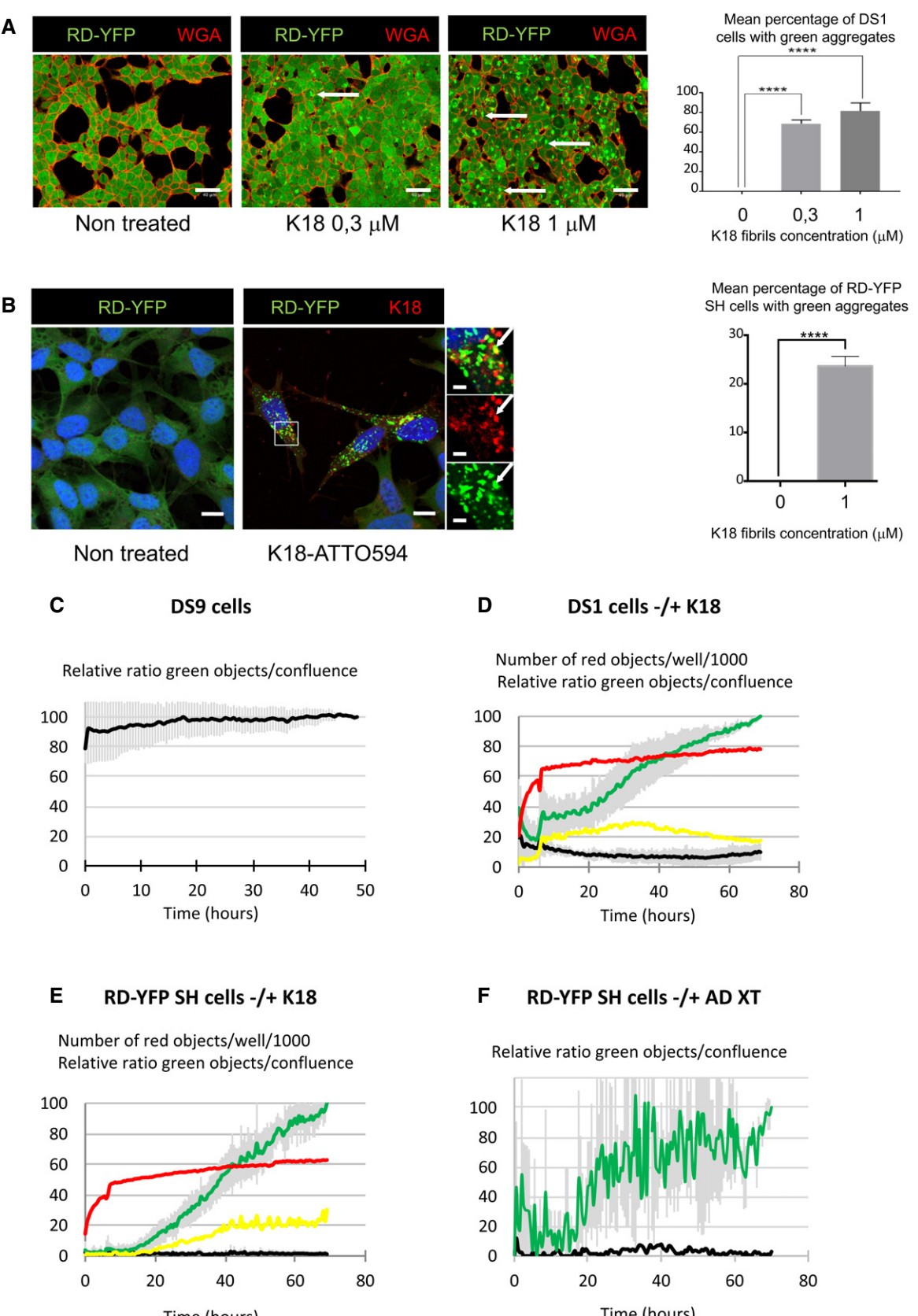

**Figure 2.**

**Figure 2. Monitoring Tau seeding.**

A  Representative confocal images of DS1 cells (expressing soluble Tau RD-YFP), non-treated and treated with increasing concentrations of non-labeled K18 fibrils, fixed after 2 days, and stained with WGA-Alexa 594 (red). The arrows point to examples of aggregate-containing cells; scale bars are 40 μm. On the right is the quantification of the mean percentage + SEM of cells where green aggregates were detected (the total number of cells analyzed over three independent experiments was 1,188 for non-treated, 1,552 for 0.3 μM, and 1,506 for 1 μM), with statistical analysis by one-way ANOVA and Bonferroni's *post hoc* test (****$P$ = 6.20E−06 and 6.74E−05 for 0–0.3 and 0–1 comparisons, respectively).

B  Representative confocal images (40×) of RD-YFP SH cells, non-treated and treated with 1 μM K18-ATTO 594 fibrils, 48 h after fibril addition. Right insets are threefold enlargements of the boxed region of treated cells, merge and single-channel pictures; white arrows point to partial colocalization. Scale bars are 10, 2 μm in the insets. On the right is the quantification of the mean percentage (24%) + SD of cells where green aggregates were detected (2,874 treated cells analyzed over six independent experiments), with statistical analysis by two-tailed unpaired *t*-test (****$P$ = 3.75E-07).

C  Relative proportion of DS9 cells expressing aggregates of Tau RD-YFP over a 50-h period monitored by IncuCyte. DS9 cells were left untreated, and the graph represents means of three independent experiments ± SD. See also Movie EV1.

D  Monitoring the conversion of DS1 cells into inclusion-containing cells over a 70-h period upon treatment with K18 fibrils. Cells were plated, treated or not with K18-ATTO 594 fibrils at 1 μM (time 0), and placed into the IncuCyte incubator for 6 h and next kept in culture for an additional 60 h after medium change. The scatter plot represents means of five independent experiments (± SD for green and black curves). See also Movie EV2.

E  Monitoring the conversion of RD-YFP-expressing SH-SY5Y cells into inclusion-containing cells over a 70-h period upon treatment with K18 fibrils. Experiment and analysis were performed as in (D), and the scatter plot shows means of three independent experiments ± SD for the green and black curves. See also Movie EV3.

F  Monitoring the conversion of RD-YFP-expressing SH-SY5Y cells into inclusion-containing cells over a 70-h period upon treatment with a cortex crude extracts from a AD-deceased patient. Experiment and analysis were performed as in (D), and the scatter plot shows means of two independent experiments ± SD for the green curve. See also Movies EV4 and EV5. IncuCyte (20× objective) was set to acquire images every 30 min, and phase-contrast and green channels (excitation 440–480 nm, 400 ms) were acquired (nine images/well). Analysis was performed with IncuCyte software as in (C–E) to give the relative proportion of cells being converted to inclusion-containing cells (the end point of treated cells was arbitrarily set at 100%, black curve: non-treated cells, green curve: treated cells).

Data information: For (C–E), IncuCyte (20× objective) was set to acquire images every 30 min, and phase-contrast and green channels (excitation 440–480 nm, 400 ms) and red (excitation 565–605 nm, 800 ms) channels were acquired (four images/well, two wells/condition). Analysis was performed with IncuCyte software to quantify the number of red objects (K18-ATTO 594 fibrils, red curve: treated cells); the relative ratio between green objects and confluence, giving the relative proportion of cells being converted to inclusion-containing cells (the end point of treated cells was arbitrarily set at 100%, black curve: non-treated cells, green curve: treated cells); the relative ratio of green and red-colocalized objects over confluence (yellow curve: treated cells).

*et al*, 2016; Swaney & Li, 2016; Keller *et al*, 2017; Sartori-Rupp *et al*, 2019)), we observed a 30% increase in K18 transfer (Fig 1C), whereas the small amount of transfer mediated by the supernatant (termed secretion) was not affected by this treatment (4.8 and 4.3% of total transfer in the absence or presence of CK666, respectively, indicated by the hatched area of the bars in Fig 1C). Inversely, when plating cocultured cells in sparse conditions, where inter-cell distance inhibits TNT growth (Abounit *et al*, 2016a; Zhu *et al*, 2018), the percentage of acceptor cells containing K18 fibrils dropped significantly (Fig 1C, right graph). As a control, we monitored in parallel cocultures the transfer of DiD-labeled vesicles from donor cells to H2B-GFP-expressing acceptor cells, which we have previously shown to be transferred predominantly through TNTs (Gousset *et al*, 2013; Abounit *et al*, 2015, 2016a; Delage *et al*, 2016). We observed similar variations in the number of acceptor cells positive for DiD or for Tau depending on the coculture conditions (CK666 treatment or sparse cells, compare Fig 1C to Fig EV1C). Overall, the absolute percentage of acceptor cells containing K18 fibrils was around 10%. Considering the internalization efficiency of K18-ATTO 594 fibrils in donor cells (which was in a range of 30–60% of cells, depending on the experiment), the transfer efficiency of the fibrils was comparable to the transfer efficiency of DiD-labeled vesicles (where labeling occurred in almost 100% of the donor cells, and around 25% of acceptor cells were positive for DiD after coculture). Together, these results suggested that both DiD-labeled vesicles and K18-ATTO 594 fibrils could be transferred between cells following similar paths considering their efficiency and their response to actin regulators. To verify that these results were not specific of the CAD model system, we repeated the experiments in human neuroblastoma SH-SY5Y cells, obtaining similar results as those with CAD cells (Fig EV1D). Overall, these results

confirmed that K18-ATTO 594 fibrils spread intercellularly *in vitro* via contact-dependent mechanisms, which were affected by conditions perturbing/increasing TNT formation. To confirm that the transfer could occur through TNTs, we analyzed whether K18-ATTO 594 fibrils were found within TNTs, identified as WGA-positive protrusions, non-adherent to the plate, and connecting distant cells (Fig 1D, compare z10 to z3, attached to the substrate). We observed that K18 fibrils could be found inside TNTs in CAD cells (arrows in Fig 1D, and orthogonal views showing that red puncta are surrounded by membrane labeled with WGA), indicating that this could be a predominant way of intercellular spreading.

**Monitoring Tau fibril seeding and propagation over cell cultures**

We next asked whether K18 Tau fibrils were able to seed new aggregates in neuronal cells after intercellular spreading. We first confirmed that K18 fibrils were able to efficiently seed aggregates in CAD cells transiently expressing full-length Tau 1N4R P301S fused to YFP (named FLTau hereafter). The appearance of FLTau aggregates was dependent on the presence of K18 fibrils (Fig EV2A and B), which partially overlapped within the cell (Fig EV2C). Then, in order to determine whether the endogenously formed FL aggregates were able to spread to neighbor cells we performed a coculture experiment, where donor FLTau-expressing CAD cells first challenged with K18 fibrils were put in contact with acceptor cells expressing H2B-mCherry for 24 h before fixation and confocal imaging. Again, as control for secretion-mediated transfer we challenged acceptor cells with the supernatant of donor cells grown in separate dishes. As shown in Fig EV2D, no transfer was observed from supernatant, whereas green dots were observed in acceptor cells in coculture conditions, and were also detected inside TNTs (see insets

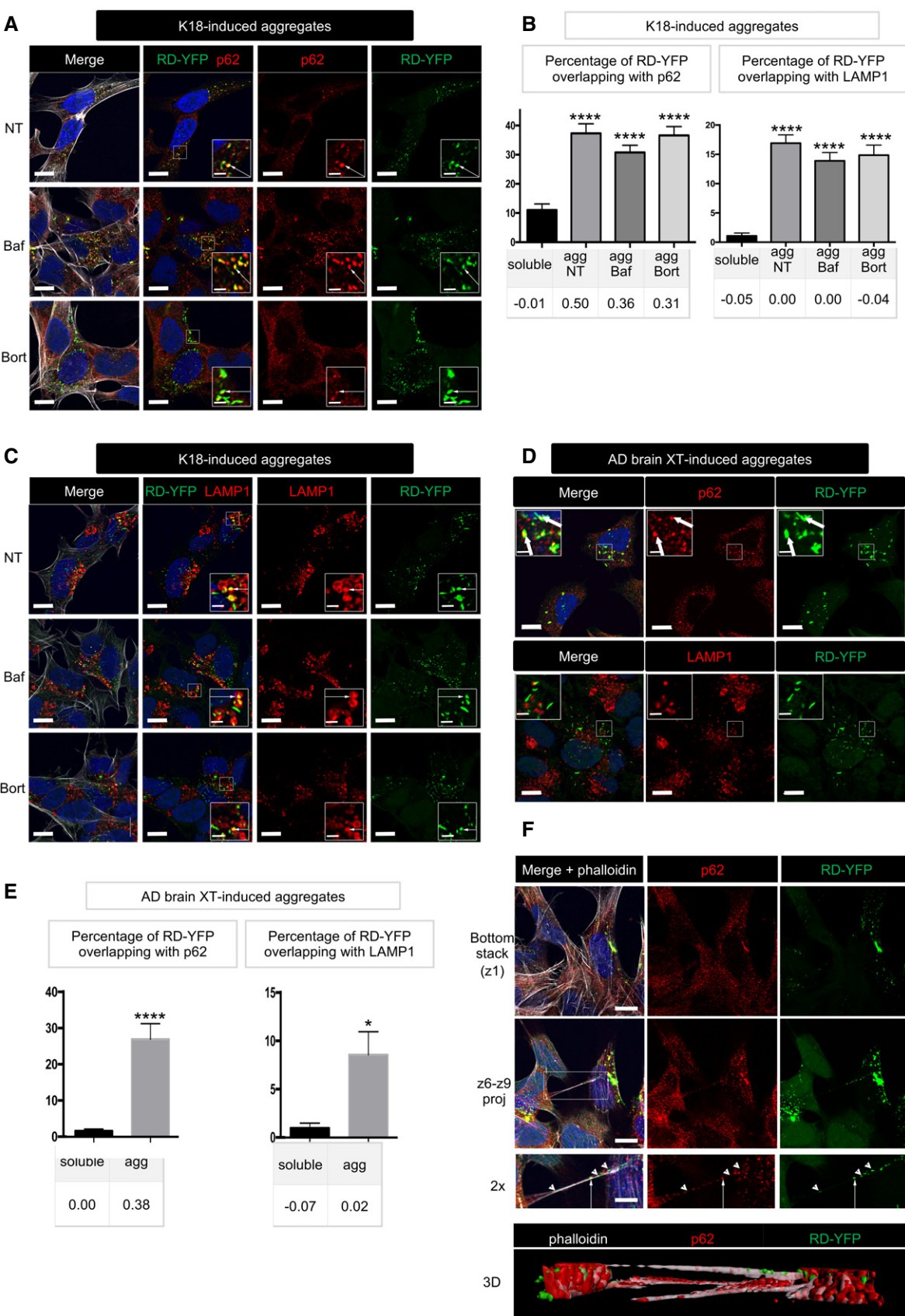

**Figure 3.**

**Figure 3. Localization of RD-YFP fibrils.**

A  RD-YFP SH cells were challenged with non-labeled K18 fibrils, incubated 2 days later with bafilomycin A1 (second line of panels), bortezomide (bottom panels), or left non-treated (NT, upper panels) for 4 h before fixation, saponin permeabilization, and staining with antibody recognizing p62 (red) and WGA (white in left panels). Images are confocal pictures after deconvolution (63×, zoom 1.8, px size = 60 nm, slice of 0.43 μm) representative of three independent experiments. Insets are threefold enlargements of the boxed regions of second column; white arrows point to colabeling; scale bars are 10, 2 μm in insets.

B  Quantification of colocalization of RD-YFP material with p62-positive structures (left graph) or LAMP1-positive vesicles (right) upon K18 fibril-induced aggregation. Confocal pictures were analyzed in 3D with Imaris software from pictures obtained as in (A), and the graphs represent the mean percentage (+ SEM) of green material overlapping with p62 or LAMP1: aggregates if cells have been converted (agg) or soluble material otherwise. Below each graph, the corresponding Pearson's correlation coefficient (PCC) is indicated. The number of cells analyzed over two independent experiments is 40, 51, 63, and 34 for each respective condition for p62, 36, 59, 44, and 27 for LAMP1. Statistically significant differences were then compared to the soluble conditions (one-way ANOVA and Tukey *post hoc* test [****$P$ = 1.33E-17, 2.57E-18, and 2.05E-20, respectively, for p62, 3.44E-16, 1.21E-13, and 1.24E-13 for LAMP1]). Note that the differences between agg NT, BafA1, and Bort were not significant neither for percentage of overlapping nor for PCC, with both p62 and LAMP1.

C  As in (A), except that anti-LAMP1 antibody was used; scale bars are 10, 2 μm in the insets.

D  RD-YFP SH cells were challenged with AD-derived brain extracts, trypsinized, and replated 3 days later for an additional 24h before fixation, saponin permeabilization, and staining with antibody recognizing p62 or LAMP1 (in red). Images are confocal pictures (63×, zoom 1.6, px size = 60 nm, slice of 0.43 μm) representative of two independent experiments. Insets are threefold enlargements of the boxed regions; white arrows point to colabeling; scale bars are 10, 2 μm in the insets.

E  Quantification of colocalization of RD-YFP material with p62-positive structures (left graph) or LAMP1-positive vesicles (right) upon AD extract-induced aggregation. Confocal pictures were analyzed in 3D with Imaris software from pictures obtained as in (D), and the graphs represent the mean percentage (+ SEM) of green material overlapping with p62 or LAMP1: aggregates if cells have been converted (agg) or soluble material otherwise. Below each graph, the corresponding Pearson's correlation coefficient (PCC) is indicated. The number of cells analyzed over two independent experiments is 21 (sol) and 27 (agg) for p62, and 12 (sol) and 17 (agg) for LAMP1. Statistically significant differences were then compared to the soluble conditions (two-tailed *t*-test, ****$P$ = 5.62E-06, *$P$ = 0.013).

F  Confocal pictures of RD-YFP cells, treated with bortezomide and labeled as in (A), except that phalloidin-Texas Red was incubated together with the secondary Alexa 647 antibody. Upper panels are the bottom slice corresponding to the substrate-attached surface of cells (z1), and below are projections of slices 6–9 of the same pictures (z6–z9, not attached to the substrate). Below is a twofold magnification of the area framed in the merged projection, showing a TNT. White arrows point to overlapping between RD-YFP aggregates and p62, and arrowheads to aggregates not colocalizing with p62 inside the TNT; scale bars are 10, 5 μm in the insets. Bottom is a 3D reconstruction (using Imaris software) of the same region.

of bottom panels). Quantitative analysis of the number of acceptor cells containing green dots (Fig EV2E) showed that FLTau transfer was strictly cell contact-dependent and on the same order of magnitude as DID transfer in these cells.

These data indicated that FLTau can be seeded by exogenously added fibrils and form endogenous FL aggregates that in turn spread mainly through a cell-to-cell contact-mediated mechanism. This is important to understand the propagation of endogenously formed fibrils; however, this system is not suitable to further investigate the underlying mechanisms. Being fluorescently labeled and recapitulating Tau aggregation, YFP-tagged Tau RD (P301L) has been previously shown to be good surrogate to investigate Tau pathology (Sanders *et al*, 2014). Thus, to follow and further investigate the seeding and transfer processes in living cells, we established a new biosensor cell line based on SH-SY5Y cells stably expressing YFP-tagged Tau RD (P301L) via lentiviral transduction (this cell line is named RD-YFP SH hereafter). We compared RD-YFP SH cells to previously established similar biosensor in non-neuronal HEK 293 cells stably expressing YFP-tagged Tau RD (LM: P301L/V337M) either in the soluble form (DS1 cells), or in the form of inherited inclusions (DS9 cells) (Sanders *et al*, 2014). In accordance with previous results (Sanders *et al*, 2014), DS1 cells were converted to inclusion-expressing cells after 2 days of exposure to the K18 fibrils, demonstrating seeded aggregation of Tau RD-YFP biosensors in a dose-dependent manner (Fig 2A). Next, we challenged the RD-YFP SH neuronal biosensor cell with the ATTO 594-labeled K18 fibrils. Also in this case, we obtained a significant proportion of RD-YFP SH cells containing green aggregates, sometimes overlapping the red fibrils (Fig 2B, and see below), confirming the ability of the K18 fibrils to seed *de novo* RD-YFP aggregates. Overall, we observed around 25% of inclusion-expressing RD-YFP SH cells 2 days after exposure to 1 μM K18 fibrils (Fig 2B). To monitor seeding and

spreading in a time-dependent manner in a quantitative assay, we took advantage of the IncuCyte-automated incubator microscope system, which allowed recording the conversion of the sensor cells upon K18 treatment. The cells were automatically imaged inside the incubator every 30 min over 3 days, and real-time quantitative live-cell and fluorescence analysis was performed. By this assay, we could quantify over time the cell confluence (bright-field analysis of the surface occupied by cells), the number of red fibrils (K18-ATTO 594 that were exogenously added), the number of green aggregates (RD-YFP, endogenously formed following fibril addition), and the number of overlapping red and green dots, which could correspond to seeding events. As a general control, we tested the IncuCyte system using DS9 cells, an established model of endogenous Tau propagation (Sanders *et al*, 2014). Our observation confirmed that these cells were able to propagate endogenous RD-YFP as aggregates over several generations (Fig 2C and Movie EV1). Our quantitative analysis in DS9 cells confirmed that the ratio of green aggregates over cell confluence remained constant over time, in accordance with the previous observation showing that in these cells the aggregated state of the Tau fibrils was stably inherited over a prolonged period (Sanders *et al*, 2014). In contrast, in DS1 or RD-YFP SH cells challenged with K18 fibrils, we observed that the number of green aggregates (reported to cell confluence) increased over time after a lag period of around 18 h (green curve in Fig 2D and E and Movies EV2 and EV3), indicating an exogenous seeding event. The discontinuity observed in all seeding curves of DS1 and RD-YFP SH cells after 6 h coincided with the medium change (allowing removal of free fibrils) that temporarily affected imaging conditions. This was not observed when monitoring DS9 cells, since these cells were not treated with fibrils and were not removed from the incubator for changing the medium (Fig 2C). Importantly, in non-treated DS1 and RD-YFP SH cells, the number of green aggregates over time was

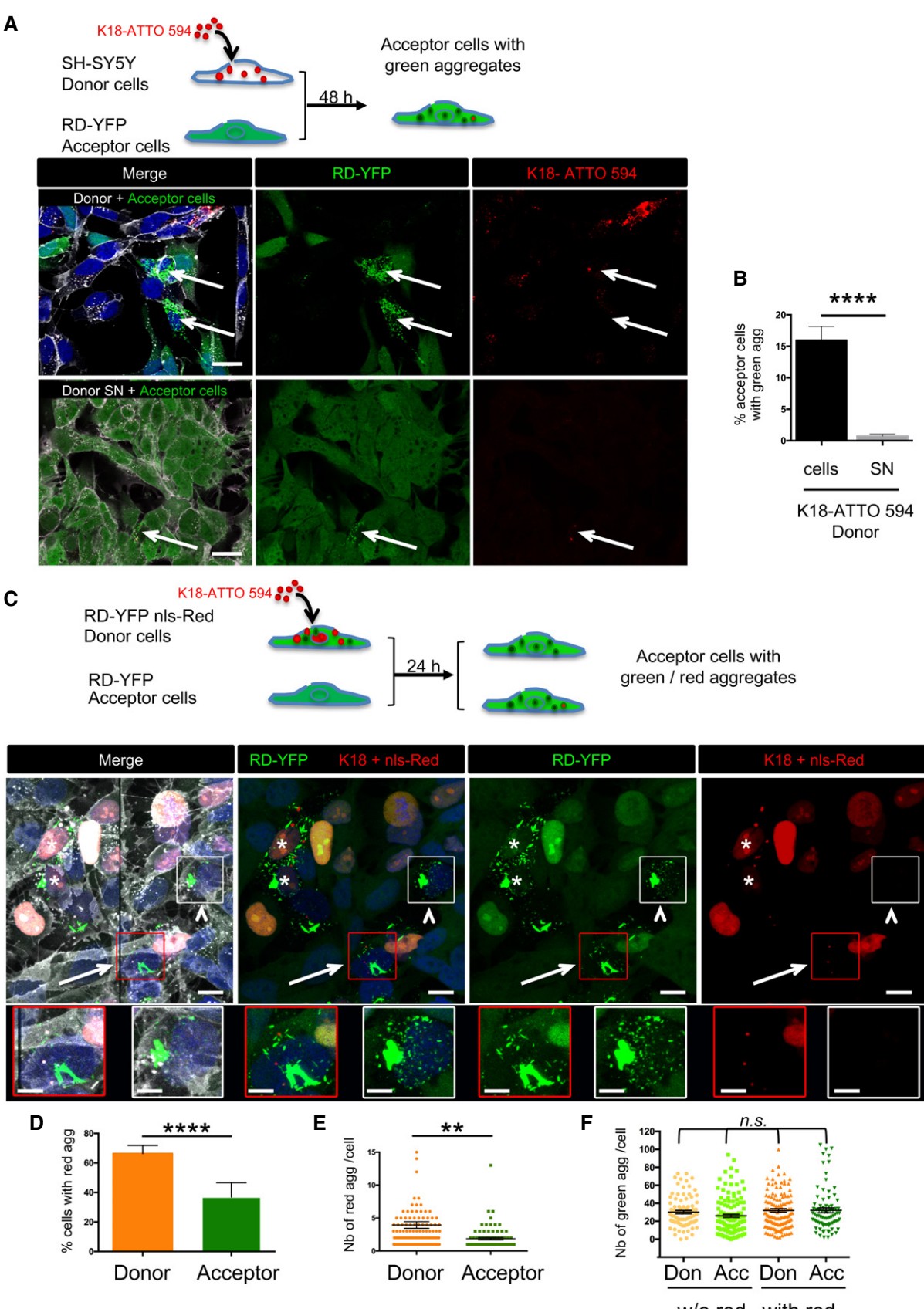

Figure 4.

**Figure 4.   Seeding of Tau after cell-to-cell contact-dependent transfer.**

A   Below the schematic representation of the experiment are representative confocal images of 2-day cocultures of RD-YFP SH cells (acceptor cells) with K18-ATTO 594 fibril-treated SH-SY5Y cells (donor cells, upper panel). Bottom panels are acceptor cells cultured with the supernatant of K18-ATTO 594 fibril-treated SH-SY5Y cells (donor SN). WGA (white) labels cell membrane and DAPI the nuclei in the merge panels. Arrows point to cells with green and red aggregates; scale bars are 20 μm.

B   Quantification of seeding after transfer in RD-YFP SH cells in the experiments described in (A), depending on the condition (donor cells or donor SN). Analysis was performed using ICY software, and data represent the number of aggregate-containing cells over the total number of green cells ± SD (the total number of RD-YFP cells analyzed over three independent experiments was 1,011 for coculture and 1,265 for SN) with statistical analysis by two-tailed unpaired *t*-test (****$P$ = 4.18E-07).

C   Below the schematic representation of the experiment are representative confocal images showing maximal intensity projections of six *z*-slices (covering 2 μm of thickness) of 24-h coculture of RD-YFP SH cells (acceptor cells) with RD-YFP SH cells expressing nls-Red and challenged with K18-ATTO 594 fibrils (donor cells). WGA (white) labels cell membrane and DAPI the nuclei in the merge panel. Stars label donor cells containing green and red aggregates, arrows indicate an acceptor cell with green and red aggregates, and the arrowheads show an acceptor cell with green aggregates but devoid of red fibrils. Below are twofold enlargement of the respective framed cells. The apparent discontinuity in the picture corresponds to the boundary between two adjacent tiles. Scale bars are 10 and 5 μm in the enlargements.

D   Analysis using ICY software of the percentage of cells containing red fibrils among the population of cells containing green aggregates. The graph shows the mean percentages with SEM (respectively, 66.3 and 36.9% for donor and acceptor cells), and statistical analysis was performed by two-tailed unpaired *t*-test (****$P$ = 3.65E−05).

E   Dot plot showing the number of red fibrils per cell with SEM among the population of cells containing green and red aggregates. Each analyzed cell is represented (note that because of the chosen scale, 6 data points are outside the axis limits), the bars indicate the means ± SEM (respectively, 3.9 and 1.9 for donor and acceptor cells), and statistical analysis was performed by two-tailed unpaired *t*-test (**$P$ = 0.0015).

F   Analysis of the number of green dots per cell among the population of cells containing green aggregates. Each analyzed cell is represented on the dot plot, and the bars indicate the means ± SEM (respectively, 30.2, 26.1, 32.0, and 32.2). Statistical analysis was performed by one-way ANOVA, and Tukey *post hoc* test and all the pairwise comparisons were not significant (n.s.).

Data information: For (D–F), counts were performed over three independent experiments using spot detector wizard under Icy software (scale 1, same threshold applied over all pictures), with a total number of analyzed cells of 222 for donor cells (136 containing red fibrils, 86 without) and 207 for acceptor cells (82 containing red fibrils, 125 without).

very low (black curve in Fig 2D–F), indicating that the appearance of aggregates was not due to random events occurring in the unseeded culture (see also Movie EV4). In K18-treated cells, the number of red puncta (represented by the red curves in Fig 2D and E) increased rapidly after challenging the cells with K18 aggregates and then reached a plateau after approximately 8 h, probably because of the limiting amount of K18 fibrils in the media. However, the relative number of overlapping green and red objects (represented by the yellow curves in Fig 2D and E), and of green objects (green curves) indicated that significant conversion of DS1 or RD-YFP SH cells into cells containing green aggregates started approximately 7 h after the signal corresponding to the K18 fibrils (red curve) reaches its plateau. If the green curve in DS1 and RD-YFP SH cells reflected only seeding of the K18-ATTO 594 on the RD-YFP SH biosensor, we would have expected to see the green curve reaching a plateau in parallel to (or some hours after) the yellow curve. Instead, the green curve kept increasing over time and plateaued after cells reached confluence (after more than 3 days). One possible explanation for this could be that seeding of new aggregates occurred also following cell division, and/or after spreading and propagation of aggregates between cells, using endogenously formed RD-YFP aggregates as templates (Fig 2D and E).

Overall, these results demonstrate that this assay is robust and allows monitoring not only Tau aggregation in real time, but also inheritance over generations of the newly formed aggregates and possibly spreading, independently of the cell type (either DS or SH cells).

To further investigate that our cell culture model was relevant to tauopathies, we used a brain extract from an AD patient (described in (Sanders *et al*, 2014)) to induce endogenous aggregation in RD-YFP cells. Movie EV4 (non-treated cells) and EV5 (treated cells) showed that seeding of RD-YFP aggregates was induced by the AD brain extract. Overall, ~50-fold more cells contained aggregates after

3 days when treated with K18 compared to AD extract. However, the kinetics of appearance of aggregates after treatment with AD extract (reported to cell confluence, Fig 2F) was very similar to that obtained after K18 fibril treatment (Fig 2E), suggesting that seeding and propagation proceeded in the same manner with both sources of fibrils, thus supporting the physiological relevance of this *in vitro* model for mechanistic studies.

## Subcellular localization and fate of endogenously formed aggregates

Next, in order to determine the intracellular localization and fate of newly formed endogenous Tau aggregates, we performed IF analysis in RD-YFP SH cultured for 2 days following the challenge with non-tagged K18 fibrils. We observed that newly formed Tau aggregates were not associated with mitochondria, early endosomes, nor Golgi structures as shown by the lack of colocalization with specific markers for these organelles (TOM20, EEA1, or Furin plus Giantin, respectively; Fig EV3A). Moreover, there was very little costaining of the aggregates with WGA, which labels all cellular membranes, including vesicles and organelles (Fig EV3A). We could also conclude that Tau aggregates were not included in aggresomes since they were not juxtanuclear, nor tubulin-positive or caged by vimentin intermediate filaments (Fig EV3A) (Johnston *et al*, 1998; Gerhardt *et al*, 2017). Altogether, these negative results suggested that most RD-YFP aggregates were either localized inside unknown compartments, or most likely they were present in the cytoplasm as free aggregates, possibly in liquid droplets, as recently proposed (Wegmann *et al*, 2018). Of interest, exogenous Tau fibrils have been reported to induce autophagy and to be present in LAMP1-positive compartments (Wu *et al*, 2013; Papadopoulos *et al*, 2017). Thus, we assessed whether endogenously formed RD-YFP aggregates were recognized as cargos for the autophagic pathway. The results

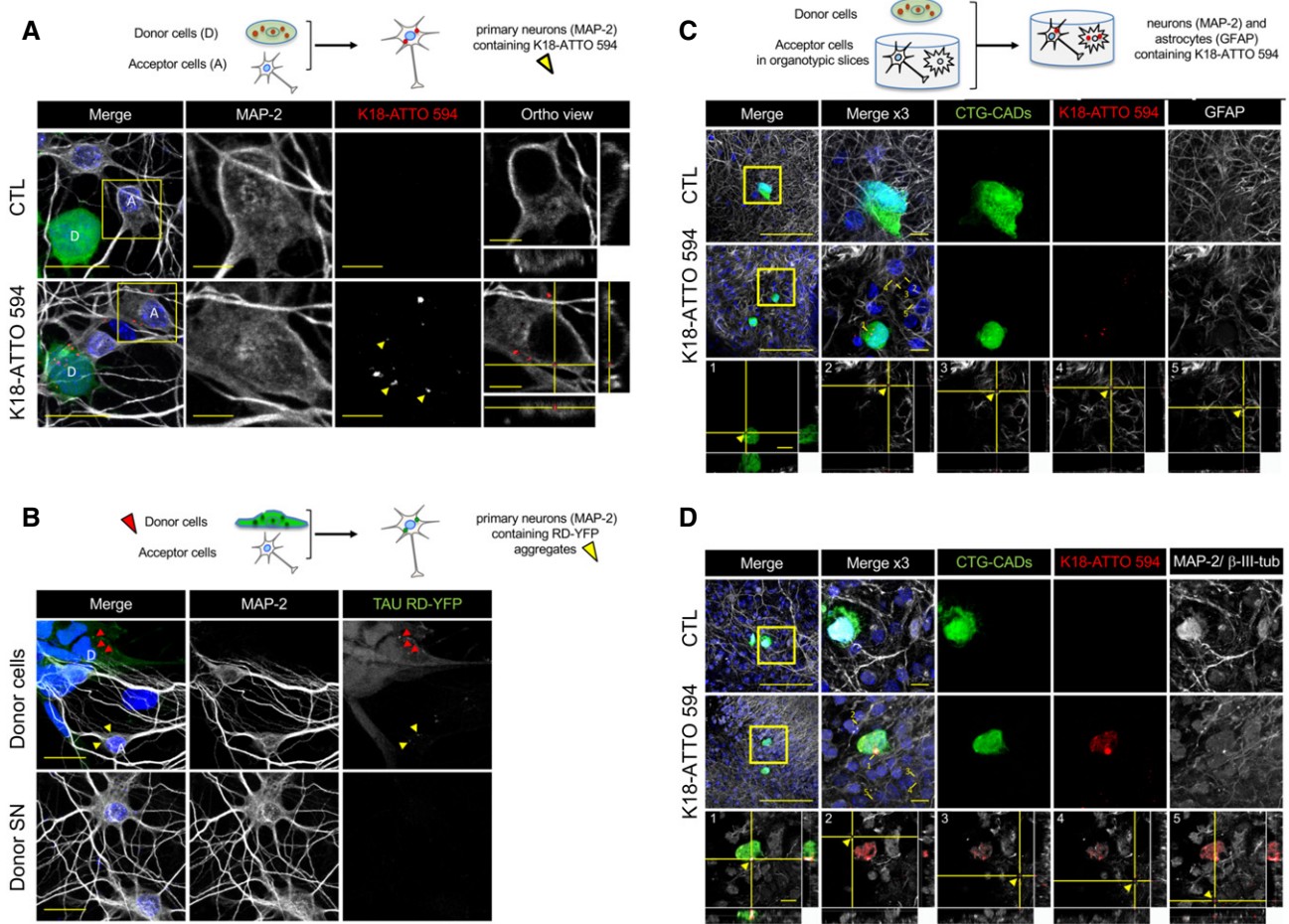

**Figure 5. Transfer of Tau fibrils to neurons and organotypic hippocampal slices.**

A   Below the schematics of the experiment are representative confocal images of donor (D) neuron-like cells (CAD cells labeled with Cell Tracker Green [CTG]) and acceptor (A) primary neurons after 24 h in coculture. The upper panels show control conditions (non-challenged CAD cells, CTL), and the bottom panels show donor CTG-CADs that were loaded with K18-ATTO 594 fibrils prior to coculture with primary cortical neurons. The images are representative Z-stack projections covering the whole cell body of donor and acceptor cells. In the merged images, white show neurons labeled with MAP-2, green are CTG-labeled donor CAD cells, red corresponds to K18-ATTO 594 puncta, and nuclei are stained in blue. The single-channel pictures are grayscale images. Yellow arrowheads point to K18 puncta inside acceptor cells. MAP-2 and K18-ATTO 594 panels are threefold enlargements of the boxed regions in the merged pictures. On the right are the orthogonal views of the same regions covering 14 slices. Scale bars are 20 μm in the merge panels, and 5 μm in the insets and orthogonal views. See also Fig EV6.

B   Below the schematics of the experiment are representative confocal images of acceptor (A) primary neurons (labeled by MAP-2) cultured with donor (D) cells, consisting of RD-YFP SH cells, first K18-challenged for 2 days, therefore expressing RD-YFP aggregates (upper panels) or with supernatant (SN) of donor cells (lower panels) after 24 h. The single-channel pictures are grayscale images. Yellow arrowheads point to Tau RD-YFP puncta inside acceptor cells, and red arrowheads point to aggregates inside donor cells. Scale bars are 20 μm.

C   Below the schematics of the coculture experiment are representative confocal images of neuron-like cells (CTG-CADs as in A) growing on top of organotypic slices (n = 6), in which the astrocytes were labeled with GFAP. The upper panels show control CTG-CADs and the middle panels show donor CTG-CADs loaded with K18-ATTO 594 fibrils, cocultured with an acceptor hippocampal slice. The images are representative Z-stack projections covering the whole cell body of donor cells. In the merged images, white are acceptor astrocytes (GFAP-positive), green are donor CTG-CADs, red are K18-ATTO 594 puncta, and nuclei are stained in blue. Insets are threefold enlargements of the boxed region in the merged picture. The orthogonal views of the bottom panels show 25 slices of K18-ATTO 594 fibrils in donor cells (1), or transfer of K18-ATTO 594 fibrils from donor CTG-CADs to hippocampal astrocytes of the organotypic slice (2–5). Yellow arrowheads point to Tau puncta inside acceptor astrocytes. Orthogonal views show each of the five K18-ATTO 594-positive puncta contained inside acceptor cells as indicated in the inset of the middle panel. Scale bars are 100 μm in the merged panel and 10 μm in the insets.

D   Same as in C, but in this case the neurons in the slices were labeled with the dendrite marker MAP-2 and the axonal marker β-III-tubulin (white), and the orthogonal views show 26 slices. Scale bars are 100 μm in the merged panel and 10 μm in the insets.

showed that RD-YFP aggregates were colocalizing with the autophagy adaptor p62 (Fig 3A). Approximately 35% of the RD-YFP aggregates colocalized with p62, with Pearson's correlation coefficient (PCC) of 0.5 (upper panels of Fig 3A and quantification in Fig 3B), while colocalization of p62 with soluble RD-YFP was not significant (Fig 3B). In addition, the RD-YFP filaments were partially colabeled with anti-ubiquitin antibody recognizing specifically K63 type of ubiquitin chains (Fig EV3B). These results suggested these endogenously formed aggregates were recognized by p62 as autophagy substrates (Cabe *et al*, 2018; Zaffagnini *et al*, 2018). We also

found that the RD-YFP-positive insoluble material was ubiquitinated (Fig EV3C), in accordance with the aggregates, or associated factors inside the same complexes, being targeted by the cells as substrates to be discarded. To further investigate this hypothesis, we analyzed whether the newly formed aggregates were delivered to lysosomes. We observed some partial overlap between RD-YFP aggregates and LAMP-1, a marker of late endosomal/lysosomal compartment (Fig 3B and C). However, the PCC was almost zero, indicating that there was no veritable colocalization with the lysosomal marker. This could be either because RD-YFP aggregates were very rapidly degraded once in the lysosomes, or because they hardly reached the lysosomal compartment. To address whether the aggregates could be degraded through the lysosomal pathway after their recruitment to autophagosomes, or whether they were degraded by proteasomes, cells were treated for 4 h with either bafilomycin A1 (an inhibitor of the late phase of autophagy) or with bortezomide (a proteasome inhibitor). No change was observed in the presence of bortezomide (Fig 3A–C). On the other hand, if autophagosomes containing RD-YFP aggregates were sent to lysosomes for degradation, we would have expected to observe an increase in their colocalization with p62 and with LAMP1 under treatment with bafilomycin A1. As expected, treatment with the autophagy inhibitor bafilomycin A1 resulted in an overall increase in the number of p62 puncta (Fig 3A). However, the percentage of colocalization with p62 or LAMP1 with respect to either the percentage of overlapping RD-YFP material or to the PCC was not affected (Fig 3A–C). We obtained similar results by labeling the cells with an antibody recognizing LC3, an autophagic membrane marker (Fig EV3D). The percentage of RD-YFP aggregates overlapping with LC3, and especially PCC, were low independently of the treatment. Overall, these results suggested that RD-YFP aggregates were recognized as autophagy cargo by p62, but likely most of the fibrils were able to escape autophagic pathway and degradation. This was consistent with the non-significant colocalization of RD-YFP aggregates with LAMP1-positive vesicles (Fig 3B and C, confirmed by LysoTracker staining in Fig EV3E). Very similar data were obtained when assaying cells at 7 or 14 days after their conversion to aggregate-containing cells, when p62 colocalization with or without bafilomycin treatment was considered (Fig EV3F and G). Overall, these data indicated that over time the autophagy pathway was not involved in fibril degradation. Further confirming the lack of degradation by either autophagy or proteasomes, we observed that the number of aggregates per cell remained stable under the different inhibitory treatments (Fig EV3H). Importantly, we obtained consistent results when cells were treated with AD brain extracts. In this case, newly formed aggregates colocalized with p62 (Fig 3D and E) and with Ub K63 (Fig EV3B) but very poorly with LAMP1 (Fig 3D and E). Overall, these data confirmed the common fate of endogenous aggregates, whether the seeding material was synthetic or naturally occurring.

Furthermore, we observed that the recruitment of Tau aggregates to autophagosomes or to other p62-positive structures was not necessary for their transport along TNTs. Indeed, we could observe TNTs (identified in IF as continuous actin-containing membranous protrusions, non-adherent to the plate, and connecting distant cells) that contained RD-YFP aggregates colocalizing or not with p62-positive compartments (see arrows and arrowheads, respectively, in the magnification of Fig 3F, and the 3D reconstruction at the bottom). This suggested that recruitment to autophagosomes or to other

p62-positive structures was not necessary for the transport of endogenously formed aggregates along TNTs. Interestingly, by assessing the diameter and the presence of actin and tubulin in aggregate-containing connections, newly formed RD-YFP Tau aggregates were found not only in classical thin TNTs containing actin but not tubulin, but also in TNT-like connections positive for tubulin (Fig EV3I). These data indicated that transfer of RD-YFP aggregates could occur through different types of TNTs (Tardivel et al, 2016; Abounit et al, 2016b; Sartori-Rupp et al, 2019).

## Seeding after spreading of endogenous fibrils by cell-to-cell contact

The above data show that endogenously formed Tau aggregates were resistant to degradation by autophagy and were able to spread between cells in a contact-dependent manner that could involve TNTs. Thus, to visualize the propagation of Tau abnormal conformation, we monitored the conversion of the Tau biosensor into aggregates in acceptor cells after the transfer of K18 fibrils from a donor cell population. To achieve this, we first challenged SH-SY5Y cells with the K18-ATTO 594 fibrils for 6 h before changing the medium. After overnight incubation, cells were trypsinized, resuspended, and cocultured with the RD-YFP SH biosensor cells (used as acceptor) in a 1:1 ratio (see schematics in Fig 4A). After 48 h, we observed that 16% of the biosensor cells were converted to green inclusion-containing cells (Fig 4A, upper panel and B for quantification). In contrast, when RD-YFP SH cells were exposed for the same period to the conditioned medium of K18-ATTO 594-treated cell culture (collected 24 h after fibril incubation, at the time coculture was started in parallel), only a very small proportion of cells contained green aggregates, indicating that only few biosensor cells were converted (0.7%, see 4A, bottom panels and Fig 4B). Consistently, no K18 fibrils could be detected by WB from 24-h conditioned medium of donor cells (Fig EV4A). These data strongly suggested that direct contact between the two cell populations favored the transfer of K18 aggregates, which in turn seeded the conversion of the biosensor. Considering that approximately 25% of donor cells contained K18-ATTO 594 fibrils after 2 days of culture and that donor and acceptor cells were plated in a 1:1 ratio, we inferred that on average one donor cell (containing K18 fibrils) could convert 0.6 acceptor cell (RD-YFP-expressing cell). This conversion efficiency was similar in magnitude to that previously observed for cell-to-cell transfer (from the results in Fig 1, we could calculate that one donor cell transferred on average to 0.3 cells after 24 h). In accordance with our kinetics data (Fig 2), the result suggests that the conversion of RD-YFP from soluble to aggregated state was not rate-limiting in relation to the spreading of the aggregates.

Altogether these data demonstrated that exogenous fibrils were transferred to neighboring cells where they could seed endogenous Tau aggregates. However, the question arose of whether endogenously formed aggregates would also be able to spread within the cell population, as it might occur in tauopathies. To address this, we made cocultures between two different RD-YFP SH populations. We used as donor cells converted RD-YFP SH cells stably harboring red fluorescent nuclei (by expression of nls-Red protein) to discriminate them from the acceptor RD-YFP SH population (see schematics of Fig 4C). Because they were exposed to seeding by K18-ATTO 594,

donor cells contained both red and green aggregates made, respectively, of the K18 exogenous fibrils and of endogenously expressed RD-YFP proteins. After 24 h of coculture, we observed that a significant portion of acceptor cells contained green aggregates and also rarely red fibrils (Fig 4C, cells shown by arrowheads and arrows, respectively, and Fig EV4B). Similar to the exogenous fibrils (Fig 4A and B), the presence of the green endogenously formed aggregates in the acceptor cells was largely dependent on cell-to-cell contact (Fig EV4B).

The green aggregates inside acceptor cells could arise either from direct transfer of green aggregates coming from donor cells, and/or from seeding in the acceptor cells of new RD-YFP aggregates using as template either red or green fibrils transferred from the donor cells. To evaluate the participation of these, possibly non-exclusive, events, we quantified the number of red fibrils and of green aggregates per cell in the donor cell population cultured alone and in the acceptor cells after coculture. After coculture, the majority of acceptor cells contained only green fibrils and only 37% of the acceptor cells contained both green aggregates and red fibrils (Fig 4D), compared with 66% in the donor cell population. Furthermore, among the 37% of acceptor cells with K18-ATTO 594 fibrils, the mean number of red dots per cell was significantly lower compared with the population of donor cells (Fig 4E, 3.9 dots/cell in donor cells and 1.9 in acceptor cells). This suggests that the transfer of red fibrils from donor to acceptor cells was quite inefficient as it only affected a small part of the acceptor cells and involved few red fibrils per cell.

Instead, in most acceptor cells (devoid of red dots), the green aggregates could either have been directly transferred from donor cells, or result from seeding by green Tau aggregates coming from donor cells. In order to evaluate the respective contribution of these events, we monitored the direct transfer of endogenously formed green aggregates by challenging donor RD-YFP SH cells with unlabeled K18 fibrils and coculturing them with SH-SY5Y (expressing H2B-mCherry) acceptor cells. As expected, transfer in these conditions occurred at a low rate (Frost *et al*, 2009a) and resulted in a very small number of green aggregates/acceptor cell (Fig EV4C). This result was similar to the number of red fibrils/cell, < 2, in the acceptor cell population shown in the coculture experiments in Fig 4C and E. In contrast, when looking at the coculture experiments of Fig 4C, the mean number of green aggregates/cell was not significantly affected in donor and in acceptor populations, and remained around 30 aggregates per cell, whether red fibrils were present or not (Fig 4F). Altogether, these results strongly suggested that the green aggregates present in acceptor cells mostly resulted from *de novo* seeding of endogenous RD-YFP. We could also conclude that green Tau aggregates that were endogenously formed in donor cells partly accounted for the seeding of new aggregates in acceptor cells after their spreading through a cell-to-cell contact-dependent mechanisms.

To further substantiate these results and confirm seeding in the absence of exogenous fibrils, we decided to make use of DS9 cells, which constitutively express insoluble RD-YFP. After 2 days of coculturing DS9 cells with RD-YFP SH cells that also expressed nls-Red, we observed that transfer of green aggregates to RD-YFP SH cells occurred predominantly in a cell contact-dependent manner (see schematics and pictures in Fig EV5A, and the respective percentage of acceptor cells with aggregates after culture with DS9,

or with DS9 supernatant in Fig EV5B). Furthermore, we could observe TNTs between DS9 cells and SH-SY5Y cells (Fig EV5C), suggesting that they could provide a way of transfer between the two cell types. Quantitation of the number of aggregates in these conditions showed that the mean number of green aggregates/cell in the acceptor population was comparable to the number of green aggregates in the same cells in the experiment of Fig 4C, suggesting occurrence of seeding after transfer (compare Fig EV5D to Fig 4F). Indeed, when monitoring only the transfer of green aggregates by coculturing DS9 cells together with SH-SY5Y cells expressing mCherry, only very few green aggregates were detected in the acceptor cells (Fig EV5E). Together, these data indicated that aggregates observed in the RD-YFP cells expressing nls-Red resulted from the conversion of endogenous RD-YFP after seeding by the native aggregates coming from DS9 cells.

### Transfer of Tau fibrils to primary neurons and organotypic hippocampal slices

To investigate whether Tau fibrils (exogenous K18-ATTO 594 labeled fibrils or endogenous RD-YFP aggregates) could be transferred intercellularly to primary neurons via a cell contact-dependent mechanism as observed in cell lines, we performed coculture experiments between CAD cells containing K18-ATTO 594 fibrils and murine primary cortical neurons for 24 h. The results in Fig 5A (bottom panels) showed that Tau fibrils (red dots) could be detected in both the donor cell population (additionally labeled with the Cell Tracker Green (CTG) dye) and in the soma and dendrites of MAP-2-positive neurons (arrowheads in the third bottom panel of Fig 5A, and see the unstacked images in Fig EV6). No red dots were detected when using control CAD cells, not loaded with K18 fibrils (Fig 5A, upper panels). Therefore, the red puncta detected in neurons were K18-ATTO 594 fibrils that have been transferred from donor CAD cells during the coculture. We performed the same type of experiment using converted RD-YFP SH cells as donor cells, and we were also able to observe specific transfer of RD-YFP aggregates to primary neurons after 96 h (Fig 5B, yellow arrowheads in the upper panels). The efficiency of transfer in these difficult experimental conditions was low, and we could not accurately quantify it; however, no transfer was detected when the supernatant of donor converted RD-YFP SH cells was applied to acceptor neurons (Fig 5B, lower panels). Thus, these data support that direct cell-to-cell contact-mediated transfer is the favored mechanism of propagation of endogenously formed Tau fibrils from neuronal cells to primary neurons.

To test the transfer of Tau fibrils in more physiological conditions, we used an *ex vivo* system consisting of organotypic hippocampal slices cocultured with K18-ATTO 594 fibril-containing CAD cells. Using this system, we could detect the transfer of K18-ATTO 594 fibrils from CAD cells that were laid on top of the slices, to GFAP-positive hippocampal astrocytes (Fig 5C) and to beta-III-tubulin and MAP-2-positive hippocampal neurons (Fig 5D), both being in the vicinity of CAD donor cells. We concluded that fibrils were inside astrocytes or neurons and not on top of them or outside the cells when cultures were imaged in orthogonal views, showing that the K18-ATTO 594 signal coincided with the cellular labeling (Fig 5A, right panels and C and D, bottom panels). Taken together, these data suggested that Tau fibrils, from exogenous origin and

endogenously formed, were able to be transferred to primary neurons and astrocytes, in a cell contact-dependent manner.

# Discussion

### An *in vitro* neuronal cell culture system that recapitulates seeding and spreading of Tau fibrils

In AD, neuronal phospho-Tau first appears in the locus coeruleus and in the trans-entorhinal cortex before it spreads to the hippocampal formation and neocortex, following a predictable pattern (Braak & Braak, 1991; Braak *et al*, 2011; Jucker & Walker, 2011; Grinberg & Heinsen, 2017; Kaufman *et al*, 2018). Other tauopathies are characterized by a distinct cellular and neuroanatomical distribution of these abnormal filaments, and by the diversity of aggregated assemblies, but eventually the misfolded protein conformation extends to larger areas of the brain, leading to specific symptoms (Frost *et al*, 2009a; Falcon *et al*, 2018). In all cases, Tau fibril assembly follows a nucleation–elongation mechanism, similar to prion-like mechanisms (Clavaguera *et al*, 2009), where the microtubule-binding domain (RD) of Tau is the core of the misassembled aggregates. The degree of tauopathy in the brain correlates with the cognitive decline in AD, suggesting that spreading of Tau deposits could be associated with disease progression (Braak & Braak, 1991; Braak *et al*, 2011). However, several questions remain to be elucidated to better understand the role of Tau in the disease. Here, we studied the mechanisms that drive the perpetuation of the amyloid conformation, the fate of the newly formed fibrils, and their cell-to-cell propagation.

Experimentally, nucleation can be obtained by external seeds of preformed Tau filaments, and this approach has been used to study seeding mechanisms *in vitro* and *ex vivo* (*in cellulo*). In order to monitor endogenous Tau aggregate generation and propagation in a time-dependent manner, we established a human neuronal model of neuroblastoma SH-SY5Y cells expressing the Tau aggregating domain RD (Frost *et al*, 2009a; Holmes *et al*, 2014; Sanders *et al*, 2014). In accordance with previous studies (Guo *et al*, 2016a; McEwan *et al*, 2017), we show that seeding is not rate-limiting and that seeded aggregation is a rapid process that occurs within 12 h after the entry of the amyloid template into the cells. Compared to other assays (Guo *et al*, 2016a; Guo *et al*, 2016b; McEwan *et al*, 2017; Chen *et al*, 2019, 2020), we were able to record in real time the cellular conversion of endogenously expressed RD upon exogenous addition of K18 fibrils, using a live-cell analysis system consisting of an automated microscope embedded within the incubator (IncuCyte System from Essen Bioscience). This allowed us to observe the kinetics of seeding, starting from the exogenous addition of fibrils to the culture and ending several days after the endogenous formation of aggregates had occurred. Interestingly, we could monitor aggregation by treating the cells with either synthetic K18 fibrils, or fibrils present in an AD-derived cortex extract (Sanders *et al*, 2014). Although the number of seeded cells was higher with the synthetic fibrils, the aggregate formation examined in living cells followed similar kinetics, no matter which seed was used, suggesting a common mechanism. These data support that this cell sensor represents a sensitive and accurate human neuronal cell system useful for evaluating the effect of drugs. In addition to monitoring the seeding process following the entry of external fibrils into

the cells, our RD-YFP SH cell model allows studying the behavior of endogenously formed aggregates. Calafate and collaborators (Calafate *et al*, 2016) have also developed mixed cultures of rat hippocampal neurons where soluble hemagglutinin (HA)-tagged TauP301L was converted to inclusion bodies when cells were grown in the presence of HEK293 cells that harbored intracellular TauP301L-GFP aggregates. Here, we also show that endogenously formed aggregates are able to propagate and seed new filaments in cell culture. Furthermore, our results extend previous findings by showing that in neuronal cells, the aggregates propagate to neighboring cells mainly through mechanisms depending on cell-to-cell contact and, importantly, can induce the formation of a second generation of aggregates. Indeed, endogenously formed aggregates can be used as templates for the formation of new aggregates over several generations and after propagation in cell culture.

Although we have shown in our cell model that aggregation of RD-Tau can be induced by both synthetic K18 fibrils and natural AD-derived fibrils, and that the resulting endogenously formed aggregates share similar kinetics of formation and cell localization, we have no formal proof that full-length Tau aggregates induced by the presence of AD-derived fibrils would behave the same regarding propagation, seeding, and fate in the cells. However, previous studies have shown that synthetic Tau fibrils and AD Tau fibrils, although having unique conformational features and differential potencies in seeding Tau aggregation, are able to induce Tau pathology when injected intracerebrally in WT or transgenic mice, or when introduced into primary neurons (Guo & Lee, 2013; Guo *et al*, 2016b; Gibbons *et al*, 2017). Therefore, we can reasonably assume that the basic mechanisms at stake for fibril entry, seeding, and spreading are the same with all types of fibrils, making our cell system a very good model for studying the common mechanisms responsible for the neoformation of aggregates from endogenous proteins induced after seeding by exogenous fibrils.

### Cell-to-cell contacts facilitate the transfer Tau fibrils to neurons and astrocytes

It has been proposed that the first Tau aggregates at the origin of tauopathies, in particular AD, are generated and accumulated inside the cells as a result of a defective cell ability to repair, degrade, and/or eliminate the misfolded proteins (Yamamoto & Simonsen, 2011; Mogk *et al*, 2018). In addition, the stereotypical pattern of Tau spreading suggests that Tau fibrils progress through the brain along neural pathways, possibly through contact-dependent mechanisms. Among those, it has been shown that Tau fibrils might induce the establishment of TNTs between primary neurons in culture and that aggregates can travel through these structures (Tardivel *et al*, 2016; Abounit *et al*, 2016b). Here, by using either exogenously added K18 fibrils, CAD cells expressing FLTau aggregates, HEK293 cells stably expressing RD-YFP aggregates (DS9 (Sanders *et al*, 2014)), or RD-YFP SH cells converted to aggregate-expressing cells as donor cells, we showed that transfer of aggregates occurs in homotypic and heterotypic cell cocultures. Importantly, similar observations were made in coculture with primary neurons, and with 3D organotypic cultures, where acceptor cells were identified to be neurons and astrocytes. Quantification showed that transfer of Tau aggregates was dependent on direct cell contact and was affected by treatments that modulate TNT formation. For example, CK-666, a

cell-permeable selective inhibitor of actin assembly mediated by Arp2/3 which increases the number of TNTs in culture (Delage *et al*, 2016; Swaney & Li, 2016; Keller *et al*, 2017; Sartori-Rupp *et al*, 2019), increased cell-to-cell transfer of K18 fibrils. Conversely, when neuronal cells were cultured in sparse conditions, which are likely to impair TNT formation, K18 fibrils transfer decreased. Most importantly, assuming that internalization, fate, and propagation could be different depending on the origin of the fibrils, here we studied the propagation of endogenously formed aggregates, either formed from FLTau or formed from RD-YFP proteins. Our data showed that RD-YFP aggregates are transferred between neuronal cells, and between neuronal cells and primary neurons in a cell contact-dependent manner. Consistently, endogenous FLTau aggregate propagation depends on direct cell contacts. Previous studies both from our group (Abounit *et al*, 2016b) and from another group (Tardivel *et al*, 2016) had detected exogenous FLTau fibrils (2N4R and 1N4R, respectively) inside TNTs of CAD and neuronal cells and TNT-like connections in primary neurons, suggesting that TNT represents a major way of propagation of FL Tau aggregates between cells. Here, we show that not only exogenous but also endogenously formed aggregates (consisting of RD-YFP or FLTau) were inside thin TNTs. Interestingly, we found RD-YFP aggregates in TNT-like connections positive for actin, and in some which were also positive for tubulin. This was not reported before; however, it could be interesting to further characterize the mechanism of transport of Tau in these protrusions to understand whether they rely on the same or different machinery (i.e., motors, organelle, or chaperones). Other studies have described Tau (monomeric or aggregated) as being secreted by an unconventional mechanism (Chai *et al*, 2012; Chen *et al*, 2020), which could be responsible for propagation of the aggregates through the culture supernatant. However, in our experimental model no K18 fibrils were detected in conditioned media and transfer of Tau aggregates (FLTau or RD-YFP) via cell supernatant (through mechanisms that could include in addition to secretion exosome production, lysosomal exocytosis, or cell death) was barely detected. Although it remains possible that mechanisms of exocytosis and endocytosis could allow transfer of fibrils between cells in close proximity, all together these data strongly suggest that in our cell models transfer through TNTs is predominant. Nevertheless, we cannot exclude the fact that FLTau spreading via secretion modes could be more relevant *in vivo* when compared to our cell culture model. Previously described anti-Tau antibodies have been shown to impair uptake, seeding, or transfer (Funk *et al*, 2015; Nobuhara *et al*, 2017; Albert *et al*, 2019; Hoskin *et al*, 2019; Weisová *et al*, 2019; Roberts *et al*, 2020). However, it is worth noting that there is no direct evidence yet showing the presence of filamentous Tau in brain interstitial fluid of patients or in mouse models, although soluble Tau could be detected in that extracellular fluid (Clavaguera *et al*, 2009; Yamada *et al*, 2011; de Calignon *et al*, 2012; Plouffe *et al*, 2012; Dujardin *et al*, 2014; Goedert, 2015). Unfortunately, although recent evidence supports the existence of TNTs *in vivo* (Korenkova *et al*, 2020; Pinto *et al*, 2020), with the current technology and lack of TNT specific markers we cannot assess at this time the presence of TNTs in brain; therefore, more studies in the future will be needed to discriminate the mechanism of Tau and other amyloid fibrils spreading *in vivo*. Interestingly, when culturing rat neurons containing endogenously expressed Tau P301L fibrils in successive microfluidic chambers, Calafate and collaborators

(Calafate *et al*, 2015) have shown that the propagation of Tau between neurons was around 5%, which is consistent with our results in SH-SY5Y cells. These authors proposed that synaptic contacts were necessary for Tau propagation; however, this could not be the case in our cell culture system, since we used undifferentiated CAD and SH-SY5Y cells, which do not form synapses. Furthermore, in our brain slice model, Tau fibrils could be transferred from non-differentiated CAD cells to differentiated neurons, which is consistent with recent work from the laboratory showing that neurons at early stage of development could form TNT-like structures (Vargas *et al*, 2019). These observations support the hypothesis that neurons are able to form functional TNT-like structures, in addition to synapses. While both ways of propagation are not mutually exclusive, it is also possible that the synaptogenic adhesion molecules used to establish the neuronal culture in Calafate *et al* (2015) influenced TNT formation in the vicinity of the synaptic bouton.

### Fate of Tau fibrils in neuronal cells

The localization and fate of Tau fibrils inside cells are still unknown. Published data suggest that exogenous Tau fibrils can enter the cells by endocytosis or by micropinocytosis, being directed to late endosomes or lysosomes (Frost *et al*, 2009b; Wu *et al*, 2013; Holmes *et al*, 2013; Calafate *et al*, 2015, 2016; Papadopoulos *et al*, 2017; Rauch *et al*, 2018). These Tau-containing vesicular structures could undergo lysosomal membrane permeabilization, thereby allowing the aggregates to escape into the cytosol and induce autophagy (Calafate *et al*, 2016; Papadopoulos *et al*, 2017; Chen *et al*, 2019). Here, we did not address the mechanisms of entry and fate of exogenous fibrils, but we rather aimed to analyze the fate of endogenously formed Tau fibrils in a model that allows monitoring the propagation of the amyloid conformation. Our results show that seeding of new filaments by fibrils transferred from other cells (in a cell contact-dependent mechanism) or by fibrils added to the extracellular medium is very efficient. In particular, by counting the endogenous and exogenous aggregates, we could demonstrate that a minimal amount of template molecules, consisting of K18 fibrils or RD-YFP aggregates, is sufficient to induce the appearance of new endogenous RD-YFP filaments. To our knowledge, this is the first time that the efficiencies of propagation and seeding have been quantitatively addressed. It is conceivable that in physiological conditions, only minute quantities of fibrils from exogenous origin could enter the cell and induce *de novo* fibrillization, resulting in an amplification of the number of aggregates. An important question that we addressed by monitoring the fate of RD-YFP aggregates is how cells cope with these endogenously formed aggregates. Overall, our data suggest that cells recognize them as species to be discarded, no matter whether initial seeding was triggered by synthetic K18 fibrils or by pathological fibrils from an AD patient. Specifically, we observed that Tau RD-YFP aggregates, or factors belonging to the same insoluble complexes, were ubiquitinylated and colocalized with the autophagy adapter p62. Consistently, they were specifically labeled by Lys 63 type of ubiquitin chains, in accordance with their recognition as autophagy cargo by p62 (Cabe *et al*, 2018; Zaffagnini *et al*, 2018). However, both proteasomal degradation and autophagy (Lee *et al*, 2013) failed to clear endogenous filaments in our neuronal cell model. Moreover, we observed

that the percentage of endogenously formed RD-YFP aggregates associated with a specific subcellular compartment (early endosomes, late endosomes or lysosomes, autophagosomes, mitochondria, Golgi apparatus) was very low. Also, Tau aggregates were not assembled in aggresomes as part of a transient response to unfolded protein stress. Recent findings suggest that Tau proteins form liquid droplets, or RNA-Tau droplets that undergo liquid–liquid phase separation (LLPS), which in turn could initiate Tau aggregation (Ambadipudi *et al*, 2017; Wegmann *et al*, 2018). Our observations are in accordance with this model, whereby most Tau fibrils would be generated and/or released in the cytoplasm.

Interestingly, Guo *et al* (2016a) have described that Tau-GFP aggregates have a great propensity to coalesce into larger aggregates over time in HEK293-derived cells. We observed the same phenomenon in DS1 cells (which originate from HEK293 cells), but not in RD-YFP SH cells, where similar proportions of small and middle-size aggregates were observed after more than 14 days of seeding with exogenous fibrils. In addition, the aggregates in HEK293 cells were shown to be recruited to autophagic vesicles without impairing autophagic flux, and were not significantly colabeled with anti-ubiquitin antibody (Guo *et al*, 2016a). In contrast, our results in RD-YFP SH cells show that some aggregates are colabeled with ubiquitin, as a subset of neurofibrillary tangles in AD brains (Cole & Timiras, 1987), and that autophagy flux is partly blocked. This is in agreement with the observed accumulation of pre-lysosomal autophagosomes in neurons of AD brains containing Tau filaments (Nixon *et al*, 2005) and with recent data using P301L-Tau-expressing mice and cells (Silva *et al*, 2019). These data support the hypothesis that cells could process Tau aggregates in different ways depending on their origin, making our neuronal cell model a very relevant model for studying tauopathies.

Altogether, our findings suggest that Tau fibril accumulation and propagation are due to their ability to seed new aggregates, to block their own degradation through the autophagic pathway, and to be directly transferred through TNTs. All these events could play key roles in the pathobiology of AD and other tauopathies.

# Materials and Methods

### Cell lines, plasmids and transfection, and reagents

Mouse neuron-like CAD (Cath.-a-differentiated) cells were described in (Gousset *et al*, 2013) and were cultured in Opti-MEM (Invitrogen) supplemented with 10% fetal bovine serum (FBS) and 1% penicillin/streptomycin (P/S). SH-SY5Y cells were a gift from Simona Paladino (Department of Molecular Medicine and Medical Biotechnology, University of Naples Federico II, Naples, Italy) and were maintained in RPMI-1640 (Euroclone), 10% FBS, P/S. DS1 and DS9 cells were described in (Sanders *et al*, 2014) and cultured in DMEM (Invitrogen), 10% FBS, P/S. All cells were trypsinized using 0.05% Trypsin-EDTA. For RD-YFP SH cells, SH-SY5Y cells were transduced with FM5-TAU RD (P301S)-eYFP lentivirus generated as described before (Tau RD; aa 244–372 of the 441 aa FL Tau 4R2N) (Sanders *et al*, 2014); next, a clonal population was obtained by limiting dilution. To obtain nls-Red lentiviral plasmid, an SV40 large T antigen nuclear localization signal (nls) peptide fused with dsRed was cloned into the shuttle vector under the control of the CMV promoter. Lentiviral shuttle plasmid, packaging plasmid, and envelope plasmid were gifts from Pierre Charneau (Pasteur Institute). HEK293T cells were cotransfected with 10 μg of the shuttle vector, 10 μg of the packaging vector, and 2.5 μg of the envelope vector, using FuGENE HD (Promega). Viral particle-containing medium was collected 48 h after the transfection and centrifuged at 1,000 *g* for 5 min to remove the cellular debris. The supernatant was concentrated on Lenti-X Concentrator (Clontech Takara), aliquoted, and stored at −80°C. nls-Red-expressing pool was obtained by lentiviral transduction in RD-YFP SH cells. Transient DNA transfections (with H2B-GFP or H2B-mCherry expression vectors, described in (Delage *et al*, 2016) or full-length Tau 1N4R P301S-YFP (Sanders *et al*, 2014)) were performed with Lipofectamine 2000 (Invitrogen) according to manufacturer's instructions. CK666 (SML0006, Sigma-Aldrich) was added at 10 μM on cells 2 h after starting coculture for 18 h, bortezomide (Focus Biomolecules) at 1 μM for 4 h before fixation, and bafilomycin A1 (Sigma) at 0.4 μM for 4 h before fixation.

### Immunofluorescence, WB, confocal microscopy, and image analysis

For imaging with the cell lines, cells were grown on glass coverslips or on Ibidi μ-dishes (Biovalley, France) for CAD cells. When no immunofluorescence was performed on the cells, cells were fixed for 15 min at 37°C in 2% PFA, 0.05% glutaraldehyde, and 0.2 M HEPES in PBS, and then additionally fixed for 15 min at 37°C in 4% PFA and 0.2 M HEPES in PBS to better preserve TNTs (Abounit *et al*, 2015). For immunofluorescence, cells were fixed with 4% paraformaldehyde and permeabilized with PBS containing 0.05% saponin before and during the incubation with appropriate antibodies. Wheat Germ Agglutinin (WGA)-Alexa Fluor© conjugates (Invitrogen) were used for membrane detection according to manufacturer's recommendation. Cell preparations were mounted in Mowiol 488. After fixation and immunostaining, images were acquired with an inverted laser scanning confocal microscope LSM700 (Zeiss). Images were acquired using the Zen acquisition software (Zeiss), deconvolution was performed with Huygens Professional software, and post-acquisition analysis was performed with ICY software (http://icy.bioimageanalysis.org/), or Imaris V7.6 (Bitplane) for quantifications of colocalization in 3D and 3D reconstruction. For counting of the number of red or green aggregates per cell, ROI was defined as cells containing green aggregates, and spot detector plugin under Icy software was used on max projection images (threshold 45, scale 2 for green, threshold 20 scale 2 for red dots).

Antibodies used for immunofluorescence were as follows: mouse anti-MAP2 (Merck Millipore, 1:500), mouse anti-β-III-tubulin (Sigma-Aldrich, 1:250), rabbit anti-GFAP (Dako, 1:500), guinea pig anti-p62 (PROGEN, 1:1,000), mouse anti-LAMP-1 (H4A3, Developmental Studies Hybridoma Bank, 1:1,000), rabbit anti-LC3 (MBL, 1:1,000), mouse anti-Ubiquitin (FK2, Enzo, 1:1,000), anti-K63-linked Ubiquitin chains (clone Apu3, Millipore, 1:200), anti α-tubulin (T9026 Sigma, 1:1,000), and Alexa-conjugated secondary antibodies (Life Technologies, 1:500). Antibodies used for Western blots were anti-ubiquitin (P4D1, Enzo 1:1,000), anti-GFP (rabbit polyclonal Invitrogen, 1:5,000), and V5 (Invitrogen, 1:1,000).

## Animals

C57BL/6 mice from our in-house colony (Institut Pasteur, Paris, France) were used. Animals were housed in cages with filter tops in a ventilated rack and maintained on food and water ad libitum. Handling of animals was performed in compliance with the guidelines of animal care set by the European Union and the French regulations on the use and care of laboratory animals, and approved by the local Pasteur Ethics Committee (CETEA; Comité d'éthique en expérimentation animale).

## Primary cells, organotypic slices, and cocultures

### Primary neuronal cultures
Primary cortical neurons were prepared from mice at embryonic day 17 as previously described (Loria *et al*, 2017). The dissociated cells were plated in 4-well dishes (45,000 cells/cm$^2$) containing poly-L-lysine-coated (1 mg/ml; Sigma-Aldrich) coverslips in Neurobasal medium (Gibco) supplemented with 10% horse serum. After 3 h, the medium was changed to complete neuronal medium (Neurobasal medium supplemented with B-27, GlutaMAX, and penicillin/streptomycin, all from Gibco). The cells were maintained at 37°C in a humidified atmosphere containing 5% $CO_2$. Five days after plating, cytosine arabinoside was added to a final concentration of 2 μM for 24 h to limit glial cell proliferation.

### Neuron-CAD coculture system
CAD cells were plated on six-well plates (37,500 cells/cm$^2$) in complete CAD medium. The following day, cells were challenged with sonicated K18-ATTO 594 fibrils (1 μM) using Lipofectamine 2000 (5 μl/nmol K18-ATTO 594) for 16 h. The next morning, donor CADs were rinsed twice with PBS and detached with trypsin. The trypsin treatment also eliminated possible K18 tau fibrils that might have remained associated with the plasma membrane, and not internalized. To clearly detect donor CAD cells and acceptor neurons, donor CADs were centrifuged and incubated with the CellTracker Green (CTG) probe (Thermo Fisher Scientific, Cat. No. C2925). Incubation was made in suspension, in a test-tube rotator (Labinco B.V.) for 30 min with CTG (12.5 μM), at 37°C with constant slow rotation. Then, donor CADs were centrifuged at 160 *g* for 10 min, the pellet was rinsed three times with complete CAD medium to remove the dye, and cells were resuspended in complete neuronal medium, seeding 60,000 cells on top of acceptor neurons (6 days *in vitro*), prepared as previously described. After 24 h in coculture, cells were washed three times with PBS, fixed with 4% PFA, immunostained, and mounted in Aqua-Poly/Mount (Polysciences). For imaging, Z-stack images covering the cell volume of donor and acceptor cells were acquired with an LSM700 inverted laser scanning confocal microscope (Zeiss), using a plan-Apochromat 63×/1.40 DIC objective.

### Neuron-RD-YFP SHSY5Y coculture
RD-YFP SH cells were plated (60,000 cells/cm$^2$) in complete medium. The following day, cells were challenged with sonicated K18 fibrils (1 μM) using Lipofectamine 2000 (5 μl/nmol K18) for 6 h before medium was replaced by complete medium. Two days after exposing RD-YFP SH cells to non-labeled K18 fibrils, when aggregates were endogenously formed in the cells, they were washed and trypsinized. After centrifugation, cells were resuspended in complete neuronal medium and 33.750 cells/cm$^2$ were laid on top of acceptor neurons (1:0.75 ratio neuron: RD-YFP). After 96 h in coculture, cells were washed, fixed, and stained as previously described. Z-stack images were acquired with the same confocal equipment as the Neuron-CAD cocultures.

### Organotypic hippocampal slice cultures
Slice preparation and culture were performed as previously described (Loria *et al*, 2017). Briefly, hippocampal slices were prepared from 6-day-old mice. After decapitation, the brain was removed and immediately submerged in cold artificial cerebrospinal fluid (87 mM NaCl, 26 mM NaHCO$_3$, 50 mM sucrose, 10 mM glucose, 2.5 mM KCl, 1.25 mM NaH$_2$PO$_4$·H$_2$O, 0.5 mM CaCl$_2$, 3 mM MgCl$_2$·6H$_2$O), bubbled with carbogen (95% O$_2$, 5% CO$_2$). Coronal slices from the hippocampus (350 μm) were cut using a vibrating-blade microtome (Leica). Slices were transferred into cell culture inserts of 0.4-μm pore size (Merk Millipore), placed in six-well culture dishes containing 1.2 ml of culture medium (50% minimum essential medium, 25% Hanks balanced salt solution [Gibco], 24% horse serum, 1% penicillin–streptomycin, supplemented with 36 mM D-glucose [Sigma-Aldrich] and 25 mM HEPES [Gibco]). Slices were incubated at 37°C in a humidified atmosphere containing 5% CO$_2$. The culture medium was replaced with fresh medium every 2 days. Cultures were maintained for 10 days *in vitro* prior to any experiment.

### Organotypic slice-CAD coculture system
CAD cells were loaded or not with 1 μM K18-ATTO 594 fibrils as indicated above. Treated (donor) and untreated (control) CAD cells were labeled in suspension with CTG following the protocol stated above. Donor and control cells were resuspended in complete CAD medium. Cells were counted, and 10 μl containing 1,000 CAD cells were added on top of each slice (acceptor). Cocultures of slice-donor CAD cells (*n* = 6) and slice-control CAD cells (*n* = 6) were incubated for 24 h and then rinsed with PBS and fixed with 4% PFA for 30 min. The samples were cryoprotected by immersion in 10% sucrose solution for 1 h and stored in 20% sucrose at 4°C until staining. Samples were freeze–thaw-permeabilized and subsequently incubated for 4 h with a blocking solution containing 20% BSA, 5% goat serum, and 0.1% Triton X-100 (all from Sigma-Aldrich) in PBS. Then, the samples were incubated during 24 h with primary antibodies and for 4 h with Alexa-conjugated secondary antibodies, and mounted using ProLong Glass (Thermo Fisher Scientific). Confocal images were acquired with a confocal microscope LSM780 (Zeiss) using a C Apo 40X/1.2W DIC III water objective. Imaging was performed with a diode 405 nm for DAPI, an argon laser line for Alexa Fluor 488, a DPSS561-nm laser for Alexa Fluor 546, and a HeNe 633-nm laser for Alexa Fluor 633.

## Preparation of fibrils, liposome-mediated transduction into cells

AD2 crude cortex extract was described in (Sanders *et al*, 2014) (fig 8C and table S2) and originated from the Brain Bank at UCSF. Tau K18 four repeats with a C-terminal V5-6His tag were synthesized and cloned into the pET24a vector using BamHI and XhoI to generate pET24-Tau K18-V5His. Plasmid DNA was transformed into *Escherichia coli* BL21 (DE3) competent cells (Invitrogen), and recombinant

colonies were selected on LB-agar plates containing 50 μg/ml kanamycin. For the large-scale expression, 2% (v/v) inoculate was used to inoculate a 50-l fermenter containing 35-l modified terrific broth media containing 1% (v/v) glycerol, 1 ml/l antifoam (DC1520), and 50 μg/ml kanamycin. The fermenter was incubated at 200 rpm and 15 l/min air. Growth was maintained at 37°C until absorbance (600 nm) measured approximately 2.0. The culture was induced with 0.1 mM isopropyl β-ᴅ-thiogalactoside (IPTG) and further incubated at 25°C overnight (20 h). The expression culture was harvested by centrifugation at 2,000 *g* for 20 min (Sorvall RC12 BP) at 4°C. The pellet was stored at −80°C. The pellet was resuspended with 4 ml/g with BugBuster® Protein Extraction Reagent (Merck 70584-4) containing benzonase (Sigma E1014-25KU) and 1 ml lysozyme (Sigma L3790). The cell suspension was stirred at room temperature for 120 min and clarified by centrifugation at 47,900 *g*, 4°C, for 90 min. Solid Imidazole was added to the lysate supernatant to a final concentration of 20 mM. Nickel Sepharose HP resin (GE Healthcare 17-5268-01) was equilibrated with buffer A (25 mM Tris base pH 8.0, 150 mM NaCl, 20 mM imidazole, and 1 ml/l protease inhibitor cocktail (Sigma P8340). The resin was loaded with the lysate supernatant and washed with ten column volume buffer A. Bound Tau protein was eluted with a linear gradient 0–100% buffer B (25 mM Tris base pH 8.0, 150 mM NaCl, 1 M imidazole, and 1 ml/l protease inhibitor cocktail) over 20 column volumes. The Tau protein was diluted threefold with 50 mM HEPES pH 7.5. Solid DTT was added to a final concentration of 5 mM, and EDTA was added to a final concentration of 2 mM and the pH adjusted to pH 7.5. The protein was further purified by ion exchange using SP Sepharose HP resin (GE Healthcare 17-1087-01) equilibrated with buffer C (50 mM HEPES pH 7.5, 50 mM NaCl, 2 mM EDTA, and 5 mM DTT). The column was washed with ten column volumes buffer C and eluted with a linear gradient 0–100% buffer D (50 mM HEPES pH 7.5, 1 M NaCl, 2 mM EDTA, and 5 mM DTT) over 20 column volumes. The purified Tau protein was dialyzed into storage buffer (PBS, pH 6.8, 100 mM ammonium acetate, and 2 mM DTT) prior to storage at −80°C. Protein identity was confirmed by peptide mass fingerprinting with 93% sequence coverage.

Fibrillation was induced by mixing K18 protein (20 μM final) and heparin (5 μM final) in storage buffer diluted twice for 24 h at 37°C under gentle shaking. Next, the mixture was ultracentrifuged at 100,000 *g* at 4°C for 40 min, the pellet was resuspended at 100 μM in PBS (considering the original concentration of monomer) for direct using or 0.1 M sodium carbonate pH 8.3 for fluorescent labeling. ATTO 594 NHS ester (Sigma) coupling was done according to manufacturer's instructions, free dye was separated from labeled fibrils by an additional ultracentrifugation, and pellet was finally resuspended in PBS at 20 μM, considering the original concentration of monomer. The final concentration used for cell transduction was 1 μM (except in Fig 2A where two doses were used). First, 2.5 μM K18 fibrils in Opti-MEM were fragmented for 5 min at RT in a VialTweeter powered by an ultrasonic processor UIS250v (250 W, 24 kHz, Hielscher Ultrasonic, Teltow, Germany) set at 80% amplitude, 5-s pulses every 2 s. Next, the K18 fibrils or AD brain extract (0.6 μg of extract in each well of a 24-well plate) were mixed with Opti-MEM containing Lipofectamine 2000 (5 μl/nmol K18) for 25 min at RT. Finally, complete medium was added (1:1), before covering the cells with this mixture. Six hours later, cell medium was replaced by complete medium.

## Flow cytometry

CAD (120 k/ml) or SH-SY5Y (100 k/ml) cells were prepared, and cocultures were performed as described in (Delage *et al*, 2016). Flow cytometry data were acquired using a CytoFLEX flow cytometer (Beckman Coulter). GFP fluorescence was analyzed at 488 nm excitation wavelength, ATTO594 and mCherry fluorescence were analyzed at 561 nm excitation wavelength, and DiD fluorescence was analyzed at 638 nm excitation wavelength. The data were analyzed using FlowJo analysis software. By subtracting the amount of transfer through secretion from the total transfer, we could infer the amount of Tau fibrils transferred via direct cell-to-cell contact. For culturing the cells in sparse conditions, the same number of cells per plate as in regular conditions was plated on a 50-cm² plate instead of a 2-cm² well, but the volume of medium was kept as low as possible (4 ml on 50-cm² plates instead of 1 ml in 2-cm² wells).

## Real-time live-cell imaging and analysis using IncuCyte

Cells were cultured on 24-well plates for 18 h (DS1 or DS9 50 k/well, RD-YFP SH 60 k/well), next treated with K18 fibrils, and immediately transferred to the incubator containing IncuCyte (Essen Bioscience) where automatic imaging of cells from four different positions per well was acquired on live cells every 30 min during 3 days (with a 20× Nikon objective). Three channels were acquired at each time point: phase channel, green channel (400-ms acquisition time), and red channel (800-ms acquisition time). A spectral unmixing was automatically applied to the pictures (3% of red signal removed from green signal). After 6 h of treatments, plates were removed between two acquisition times, and medium was replaced by fresh complete medium. Data were then analyzed using IncuCyte ZOOM software.

## Statistical analysis

Statistical analyses were performed using Prism 6.0 (GraphPad) as indicated for each experiment.

# Data availability

This study includes no data deposited in external repositories.

**Expanded View** for this article is available online.

## Acknowledgements

We acknowledge the Center for Translational Science (CRT)—Cytometry and Biomarkers Unit of Technology and Service (CB UTechS), the Image Analysis Hub, and the UTechS Bioimagerie photonique (PBI) at Institut Pasteur for their support in conducting this study. We are grateful for financial support to C.Z. from JPND Research, EU Joint Program 2014 "NeuTARGETs", ANR (Agence Nationale pour la Recherche) "NeuroTUNN" 2016, Fondation Vaincre Alzheimer LECMA, and France Alzheimer and equipe FRM (Fondation pour la Recherche Medicale) 2014. FL was recipient of Marie Skłodowska-Curie individual fellowship (Neurotunn 702465) and next was funded through the Marie Skłodowska-Curie Action COFUND 2015 (EU project 713366—InterTalentum).

**The paper explained**

**Problem**

In Alzheimer disease (AD) as in other tauopathies, Tau neurofibrillary tangles are able to seed abnormal conformations on normal proteins, initiating a self-amplifying cascade, and can spread from their initial production site to other areas in the brain, following well-defined pathways. However, the underlying events that explain these features are still a matter of debate, as the fate and behavior of endogenously formed aggregates have not been assessed.

**Results**

We established a neuronal reporter cell system, allowing to record in real time the conversion of endogenously expressed Tau RD domain from soluble state to aggregates upon addition of external fibrils, either synthetic or from AD patient brain extract. We show that seeding is not rate-limiting and follows the same kinetics with both sources of fibrils. Further, the endogenously formed aggregates are recognized as autophagic cargoes, but fail to be transferred to lysosome for degradation as the autophagy flux is partly blocked. Both exogenous and endogenous aggregates, including those composed of full-length Tau, are transmitted between cells in a contact-dependent manner and found inside TNTs in neuronal cell lines. Tau fibrils are also transferred through direct cell contact to primary neurons and 3D organotypic cultures, where recipient cells were identified to be neurons and/or astrocytes.

**Impact**

This work gives an original and comprehensive picture of the pathobiology of AD and other tauopathies by analyzing the intracellular events that lead to the formation and spreading of Tau aggregates. It provides the groundwork for future intervention therapies specifically designed to improve clearing of Tau fibrils and to block their propagation throughout the brain.

## Author contributions

CZ conceived the project; CZ and CB conceptualized the data; PC, FL, JYV, and CB involved in methodology; PC, FL, JYV, JT, and CB involved in formal analysis and investigated the data; FL, CB, and CZ wrote the original article and drafted the manuscript; FL, JYV, MID, GO, CB, and CZ wrote, reviewed, and edited the manuscript; CZ acquired funding; CZ, MD, and GO provided resources; and CB and CZ supervised the data.

## Conflict of interest

The authors declare that they have no conflict of interest.

## For more information

https://research.pasteur.fr/en/team/membrane-traffic-and-pathogenesis/
https://www.pasteur.fr/en/research-journal/reports/alzheimer-s-disease-towards-new-diagnostic-and-therapeutic-tracks
https://www.alzforum.org
https://www.alzforum.org/alzpedia

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
