## [Review Process File · EMBO Molecular Medicine]

Fate and propagation of endogenously formed Tau aggregates in neuronal cells

Patricia Chastagner, Frida Loria, Jessica Vargas, Josh Tois, Marc Diamond, George Okafo, Christel Brou, and Chiara Zurzolo

DOI: [10.15252/emmm.202012025](https://doi.org/10.15252/emmm.202012025)

Corresponding authors: Chiara Zurzolo (chiara.zurzolo@pasteur.fr)

Review Timeline:

Submission Date:	14th Jan 20
Editorial Decision:	6th Feb 20
Revision Received:	29th May 20
Editorial Decision:	22nd Jun 20
Revision Received:	6th Aug 20
Editorial Decision:	24th Aug 20
Revision Received:	18th Sep 20
Accepted:	21st Sep 20

Editor: Jingyi Hou

Transaction Report:

6th Feb 2020

Dear Prof. Zurzolo,

Thank you for the submission of your manuscript to EMBO Molecular Medicine. We have now heard back from the three referees whom we asked to evaluate your manuscript. As you will see from the reports below, the referees acknowledge the potential interest of the study. However, they also raised a series of concerns about your work, which should be convincingly addressed in a major revision of the present manuscript. Without repeating all the points raised in the reviews below, efforts should be made to:

- provide data using more physiologically relevant tau fibrils to make the study more conclusive, as commented by referee #3
- provide in vivo data to enhance the medical relevance of the findings, as commented by referee #2.

During our pre-decision cross-commenting process (in which the referees are given the chance to make additional comments, including on each other's reports), referee #2 said "About the relevance of including more physiological relevant data, as suggested by reviewer 3, it will be important to test AD derived tau fibrils, Probably, the authors could address that point in a revised manuscript."

All other issues raised by the referees need to be satisfactorily addressed as well. We would welcome the submission of a revised version within three months. Please note that EMBO Molecular Medicine strongly supports a single round of revision and that, as acceptance or rejection of the manuscript will depend on another round of review, your responses should be as complete as possible.

Please also contact us as soon as possible if similar work is published elsewhere. If other work is published, we may not be able to extend the revision period beyond three months.

I look forward to receiving your revised manuscript.

Yours sincerely,
Jingyi Hou

Jingyi Hou

*** Instructions to submit your revised manuscript ***

**** PLEASE NOTE **** As part of the EMBO Publications transparent editorial process initiative (see our Editorial at <https://www.embopress.org/doi/pdf/10.1002/emmm.201000094>), EMBO Molecular Medicine will publish online a Review Process File to accompany accepted manuscripts.

To submit your manuscript, please follow this link:

Link Not Available

- 1) a .doc formatted version of the manuscript text (including Figure legends and tables). Please make sure that the changes are highlighted to be clearly visible to referees and editors alike.
- 2) separate figure files*
- 3) supplemental information as Expanded View and/or Appendix. Please carefully check the authors guidelines for formatting Expanded view and Appendix figures and tables at <https://www.embopress.org/page/journal/17574684/authorguide#expandedview>
- 4) a letter INCLUDING the reviewers' reports and your detailed responses to their comments (as Word file)

Also, and to save some time should your paper be accepted, please read below for additional information regarding some features of our research articles:

- 5) The paper explained: EMBO Molecular Medicine articles are accompanied by a summary of the articles to emphasize the major findings in the paper and their medical implications for the non-specialist reader. Please provide a draft summary of your article highlighting
 - the medical issue you are addressing,
 - the results obtained and
 - their clinical impact.

6) For more information: There is space at the end of each article to list relevant web links for further consultation by our readers. Could you identify some relevant ones and provide such information as well? Some examples are patient associations, relevant databases, OMIM/proteins/genes links, author's websites, etc...

7) Author contributions: the contribution of every author must be detailed in a separate section (before the acknowledgments).

8) EMBO Molecular Medicine now requires a complete author checklist (<https://www.embopress.org/page/journal/17574684/authorguide>) to be submitted with all revised manuscripts. Please use the checklist as a guideline for the sort of information we need WITHIN the manuscript as well as in the checklist. This is particularly important for animal reporting, antibody dilutions (missing) and exact p-values and n that should be indicated instead of a range.

9) Every published paper now includes a 'Synopsis' to further enhance discoverability. Synopses are displayed on the journal webpage and are freely accessible to all readers. They include a short stand first (maximum of 300 characters, including space) as well as 2-5 one sentence bullet points that summarise the paper. Please write the bullet points to summarise the key NEW findings. They should be designed to be complementary to the abstract - i.e. not repeat the same text. We encourage inclusion of key acronyms and quantitative information (maximum of 30 words / bullet point). Please use the passive voice. Please attach these in a separate file or send them by email, we will incorporate them accordingly.

You are also welcome to suggest a striking image or visual abstract to illustrate your article. If you do please provide a jpeg file 550 px-wide x 400-px high.

10) A Conflict of Interest statement should be provided in the main text

11) Please note that we now mandate that all corresponding authors list an ORCID digital identifier. This takes <90 seconds to complete. We encourage all authors to supply an ORCID identifier, which will be linked to their name for unambiguous name identification.

Currently, our records indicate that the ORCID for your account is 0000-0001-6048-6602.

Please click the link below to modify this ORCID:
Link Not Available

12) The system will prompt you to fill in your funding and payment information. This will allow Wiley to send you a quote for the article processing charge (APC) in case of acceptance. This quote takes into account any reduction or fee waivers that you may be eligible for. Authors do not need to pay any fees before their manuscript is accepted and transferred to our publisher.

Photos 400-800 DPI

*Additional important information regarding figures and illustrations can be found at <http://bit.ly/EMBOPressFigurePreparationGuideline>

***** Reviewer's comments *****

Referee #1 (Comments on Novelty/Model System for Author):

K18 coculture experiment: transfer is likely dependent on local concentration at the cell surface; this is obviously going to be higher when cells are in close proximity than when diluted by media or by sparse conditions.

It's puzzling that the K18 aggregates don't overlap with the full length tau p301L inclusions, but the kinetics experiment is well done and provides interesting, if not always easily understandable, data.

Cell contact vs uptake of exogenous tau species is a very important point - from a therapeutics perspective, if the tau has to stay within a cell, antibody treatment to block uptake seems unlikely. Yet numerous groups have used antibodies to at least partially block uptake/transfer. Thus, both mechanisms might be at play. Nonetheless, given the clinical relevance of this issue, an additional experiment in which transfer of full length misfolded tau between adjacent neurons is observed in the presence of a well characterized tau antibody would be welcome

Referee #1 (Remarks for Author):

K18 coculture experiment: transfer is likely dependent on local concentration at the cell surface; this is obviously going to be higher when cells are in close proximity than when diluted by media or by sparse conditions.

It's puzzling that the K18 aggregates don't overlap with the full length tau p301L inclusions, but the kinetics experiment is well done and provides interesting, if not always easily understandable, data.

Cell contact vs uptake of exogenous tau species is a very important point - from a therapeutics perspective, if the tau has to stay within a cell, antibody treatment to block uptake seems unlikely. Yet numerous groups have used antibodies to at least partially block uptake/transfer. Thus, both

mechanisms might be at play. Nonetheless, given the clinical relevance of this issue, an additional experiment in which transfer of full length misfolded tau between adjacent neurons is observed in the presence of a well characterized tau antibody would be welcome

Referee #2 (Remarks for Author):

In this work, Chastagner et al. have analyzed the transfer of tau aggregates through a direct cell-cell contact, involving the presence of tunneling nanotubes for that cell-to-cell transfer of the aggregates. Mainly, the work has been carried out using cell culture analysis. The technical quality of the work is high but the extrapolation of the data obtained in the cell cultures to medical purposes is not clear.

Specific points:

- TNT is a mechanism of cell to cell transfer of proteins, or other compounds. Cytoskeletal components of nanotubes include actin, but also (depending of caliber size of TNT) microtubules could be also present (Vidulescu C. et al 2004, J Cell Mol. Med. 8, 388; Davis D. and Sowiski S. 2008, Nature rev. Mol Cell Biol. 9, 431). In figure 1D, a WGA-positive TNT is shown, but it can be advisable to indicate the caliber diameter size to know if microtubule protein (including tau) may be present.
- The work has been focused on the role cell-cell contact and little is commented about other ways for tau transfer from cell to cell. In this area, different analyses have been extraordinarily focused only in one event, and sometimes there are some controversies from work to work. For example, at the same time of reading the present manuscript, I also read Chen X. et al. 2020 Mol Neurod 6:15 or Ugbode C et al. 2019 J.Biol.Chem 294, 18967. In those works, the role of autophagy lysosomes or endosomes appear to be important, but in the present work, it is indicated that tau aggregates are essentially not associated to endosomes, lysosomes, ... This point should be further explained to indicate common or different ways for tau propagation.
- It was indicated that tau deposits co-labeled with ubiquitin. Thus, it will be of interest if ubiquitin is covalently bound to tau protein, or not. This analysis could be carried out by WB, and it could suggest if more than one ubiquitin molecule could be bound to tau. It was shown that tau aggregates were resistant to degradation by autophagy, but little was discussed about proteasome degradation.
- In the first paragraph of the Discussion, the previous and recent works of Braak should be quoted. In the recent work of Braak is suggested that the first tau inclusions are present at the locus coeruleus.
- Although TNTs were first described (see ref 43) in "neuronal-like" cultured-PC-12 cells, little is known about their presence (and physiological or pathological role), in human brain and in tauopathies like Alzheimer disease. In this work, the analyses have been mainly carried out in cell culture but not in any "in vivo model".

In summary, the technical quality of the analyses done is high but the complementation of the data with "in vivo" studies will greatly improve the meaning of the work.

Referee #3 (Comments on Novelty/Model System for Author):

They have developed a nice model system. It would be important to evaluate the efficacy of the system using AD derived tau fibrils.

Referee #3 (Remarks for Author):

EMM-2020-12025 Fate and propagation of endogenously formed Tau aggregates in neuronal cells

The goal of the study by Chastagner et al is to develop a neuronal model capable of tracking the fate of an exogenously provided tau fibril- starting from getting internalized by the cell, to aggregating and seeding endogenous tau proteins, lastly followed by its spread to a neighboring cell via tunneling nanotubes (TNT). The authors also show that tau aggregates, although colocalizes with p62, do not undergo clearance via lysosomal pathway instead, they discard this unwanted aggregated protein principally through TNT, resulting in its spreading. Finally, they culminate their study by verifying their cell to cell contact theory in two coculture system involving tau fibril containing CAD cells as donors to MAP2 positive neuronal cultures and hippocampal organotrophic cultures respectively.

The following study provides novel insights into how tau fibril can aggregate, seed and transfer its toxicity to neighboring cells in a monoculture system.

Comments:

- 1) The most important outreach of the model proposed in this study is in its applicability towards understanding toxic tau propagation in diseases like Alzheimer's (AD). For a while it has been known that heparin induced tau aggregation do not form AD specific tau aggregates (or aggregates resembling any other tauopathy). Hence, the authors need to use a more physiologically relevant tau fibril in their model (eg- tau fibril extracted from AD brain).
- 2) Since the authors used a synthetic tau fibril seed there is a possibility that other modes of transmission such as exosomes gets masked due to lack of a physiologically potent and antigenic tau fibril capable of being identified and hence loaded into an exosomal vesicle. This may explain why cells in their model preferentially uses TNT (cell to cell contact) over other non-cell to cell contact methods (this may be an artifact of their culture system caused by the choice of their toxic trigger). The authors need to re-evaluate the properties of their neuronal model system as they change their exogenous tau fibril source.

1. Editor main comments:

Without repeating all the points raised in the reviews below, efforts should be made to:

- provide data using more physiologically relevant tau fibrils to make the study more conclusive, as commented by referee #3
- provide in vivo data to enhance the medical relevance of the findings, as commented by referee #2.

During our pre-decision cross-commenting process (in which the referees are given the chance to make additional comments, including on each other's reports), referee #2 said "About the relevance of including more physiological relevant data, as suggested by reviewer 3, it will be important to test AD derived tau fibrils, Probably, the authors could address that point in a revised manuscript."

We agree with the editors and reviewers that we needed to further validate our model by using natural tau fibrils on one hand, and by showing that full-length Tau is seeded similarly to RD-YFP Tau by exogenous fibrils. We performed experiments to address these two points, described in detail in the specific answers to the reviewers. Specifically, we were able to show that AD-derived Tau fibrils induce the formation of RD-YFP aggregates following a kinetics similar to that induced by K18 fibrils (**see new fig 2F and new movies EV4, 5**). In addition, these seeded aggregates have the same cellular fate as the ones induced by K18 fibrils: they are ubiquitinated and able to recruit p62, therefore recognized as autophagy cargo, but fail to reach lysosomes for complete degradation (**see new fig 3D, E and fig EV3B**). Thus, these new results show that our RD-YFP-expressing cell model is relevant and could be used to study the formation of aggregates.

All other issues raised by the referees need to be satisfactorily addressed as well.
See the specific comments below.

***** Reviewer's comments *****

Referee #1 (Comments on Novelty/Model System for Author) and Remarks for Author:

2. K18 coculture experiment: transfer is likely dependent on local concentration at the cell surface; this is obviously going to be higher when cells are in close proximity than when diluted by media or by sparse conditions.

We thank the reviewer for this comment, which is very relevant. First, we want to highlight the fact that cells are fully trypsinized before plating them in coculture, thus eliminating the fibrils that could be present at the cell surface of donor cells. We also made secretion controls, where the medium of donor cells cultured in the exact same conditions as in the coculture was used as potential source of fibrils, and put back on acceptor cells with a resulting minimal amount of transfer. In these latter conditions, the local concentration of fibrils at the cell surface (produced by donor cells) should be the same as in coculture conditions. Nonetheless, we totally agree with the reviewer that medium dilution could change the efficiency of transfer, if it was dependent on local secretion mechanisms. To this aim we minimized (as much as possible) the volume of medium when performing transfer experiments in sparse cell conditions (**the precise description of the protocol was added in material and methods, page 11, lines 13-15**). On the other hand, we believe that the experiments using CK666 (fig 1C and

EV 1C), a compound known to increase the number of TNTs (Delage et al, Sci Rep 6: 39632, 2016, Swaney et al, Curr Opin Cell Biol 42:63, 2016, Keller et al, Invest Ophthalmol Vis Sci 58:5298, 2017, Sartori-Rupp et al, Nat Comm 2019), are complementary and clearly indicate that affecting the formation of TNTs affects transfer. Indeed, we observed an increase in K18 or DiD transfer proportional to the increase in TNT number. If secretion was only at stake, transfer would not be affected by CK666 treatment. Together with the visualization of aggregates inside TNTs (Fig 1D, 3D and **new fig EV3I**), we think that the hypothesis of transfer through TNTs is well consolidated in this work. Although it remains possible that mechanisms of exocytosis and endocytosis between cells in close proximity could happen and allow transfer in a cell-contact-independent manner, we believe that it is not a major event in our experimental conditions. We have nonetheless discussed that we could not exclude this possibility in the discussion (**page 28, lines 2-9**).

3. It's puzzling that the K18 aggregates don't overlap with the full-length tau p301L inclusions, but the kinetics experiment is well done and provides interesting, if not always easily understandable, data.

We would like to point out that we obtained the kinetic data of figure 2 using the RD-YFP construct, and not the full-length (FL) Tau P301S. Figure 2B shows that RD-YFP aggregates partially overlap with K18 red fibrils. We did not expect this colocalization to be very high, since it is likely that seeding is only initiated by K18, and the endogenously formed aggregates self-propagate by secondary seeding.

Furthermore, following the remark of this reviewer, we now have added a new panel (**new fig EV2C**) showing colocalization of K18 with FL aggregates, further confirming that K18 fibrils were able to seed full-length Tau inclusions. Interestingly, the K18 fibrils seem to be embedded into the FL Tau tangle (as in the example shown in **fig EV2C**). This observation supports K18 fibrils being the starting point of seeding and then becoming trapped into the growing body of newly formed Tau aggregates.

4. Cell contact vs uptake of exogenous tau species is a very important point - from a therapeutics perspective, if the tau has to stay within a cell, antibody treatment to block uptake seems unlikely. Yet numerous groups have used antibodies to at least partially block uptake/transfer. Thus, both mechanisms might be at play. Nonetheless, given the clinical relevance of this issue, an additional experiment in which transfer of full length misfolded tau between adjacent neurons is observed in the presence of a well characterized tau antibody would be welcome.

We agree with the reviewer that the therapeutic use of antibodies was tested in several recent papers, based on the assumption that clearing the soluble extracellular Tau responsible for propagation, or removing the pathological forms of Tau, could decrease the clinical progression of AD. However, although several Tau passive immunotherapy programs have been launched (Pedersen & Sigurdsson Trends Mol Med 21:394 2015, and Hoskin et al, Exp Opin Investig Drugs 28: 545,2019), no optimized Tau immunotherapy strategy is disclosed yet. Of note, most of the described efficient antibodies (like 6C5 from Nobuhara et al, Am J Pathol 187: 1399, 2017, or RG7345, see Hoskin et al, 2019, or antibody D used in Courade et al, Acta Neuropathologica 136:729, 2018 and in Albert et al, Brain, 142:1736, 2019), target domains or post-translational modifications outside the RD domain that we use, and are therefore

would not be useful in our conditions. On the other hand, antibodies targeting the RD domain (DC8E8 of Weisová et al, *Acta Neuropathol. Commun* 7:129 2019, or E2814 of Roberts et al, *Acta Neuropathol. Commun* 8:13, 2020), most probably impair entry and/or seeding of pathogenic Tau (either from AD or K18), and might be more relevant to addressing cell-to-cell transfer in our model. To this aim we first assessed whether in our conditions K18 fibrils were released into the cell culture medium. Thus, we partially purified and concentrated insoluble material by ultracentrifugation from supernatants of cells 24 hours after challenging them with K18 fibrils, and tested these fractions for the presence of K18 fibrils by WB (**new fig EV4A**). We were unable to detect immunoreactive material in the cell supernatants. This indicates that fibrils are not secreted in detectable amounts in the medium in our experimental conditions, supporting our hypothesis that they propagate mainly through TNTs without any contact with the extracellular medium.

In addition, we would like to stress that in our cell models, which include CAD and SH-SY5Y cell lines, but also primary neurons and astrocytes, K18 Tau fibrils were unable to enter the cells spontaneously. Since we made use of a protein carrier (lipofectamine) to facilitate entry of K18 into the cells, in accordance with the work of Nobuhara et al *J Pathol* 187:1399, 2017), K18 fibril treatment monitors the seeding alone, and not internalization and seeding. Therefore, all considering, using antibodies described to impair entry of fibrils, including 4R-made, like DC8E8 described in Weisová et al (2019), would not be relevant in our experimental conditions.

As for the secondary seeding, resulting from the propagation of K18 fibrils or endogenously formed RD-YFP aggregates, we found that transfer mainly occurs through a cell-contact dependent mechanism (see figures 4, EV4 and EV5). If transfer occurred through secretion mechanisms, aggregates should be either released freely into the medium, and therefore (in our hands) unable to penetrate another cell, or included into vesicles that would be internalized by the neighboring cells.

Notwithstanding, we agree with the reviewer that experiments monitoring transfer of FL Tau aggregates would be necessary to completely address this point. Since we cannot completely rule out that part of the cell-to-cell transfer involves cell contact-independent mechanisms, we modified the discussion and included the mention of possible antibody treatment (**page 28, lines 1-11**). See also point 7.

Referee #2 (Remarks for Author):

5. In this work, Chastagner et al. have analyzed the transfer of tau aggregates through a direct cell-cell contact, involving the presence of tunneling nanotubes for that cell-to-cell transfer of the aggregates. Mainly, the work has been carried out using cell culture analysis. The technical quality of the work is high but the extrapolation of the data obtained in the cell cultures to medical purposes is not clear.

We thank the reviewer for this comment, and we agree with him/her that our work, although aimed at understanding the molecular mechanisms of propagation and the fate of Tau aggregates within the cells, lacked strong evidence showing that our cell model is physiologically relevant. We have now added data using AD-derived Tau fibrils from brain extracts, showing that RD-YFP cells are converted to aggregate-expressing cells similarly as under K18 treatment (**new fig 2F, 3D, E and fig EV3B, and movies EV4 and EV5**). In our opinion

these new results have increased both the quality of the paper and the relevance of our model for further (more applied) studies.

Specific points:

6. TNT is a mechanism of cell to cell transfer of proteins, or other compounds. Cytoskeletal components of nanotubes include actin, but also (depending of caliber size of TNT) microtubules could be also present (Vidulescu C. et al 2004, J Cell Mol. Med. 8, 388; Davis D. and Sowiski S. 2008, Nature rev. Mol Cell Biol. 9, 431). In figure 1D, a WGA-positive TNT is shown, but it can be advisable to indicate the caliber diameter size to know if microtubule protein (including tau) may be present.

We agree with the reviewer, that TNT diameter could be diverse, although “classical” TNTs have been recently described in the cell lines that we used for our study (CAD and SH-SY5Y), to be composed of a single or various iTNTs, devoid of microtubules (Sartori-Rupp et al, Nat Comm 10:342,2019). Following this reviewer’s suggestion, we performed new experiments to detect actin (by Phalloidin labeling) and microtubules in addition to RD-YFP aggregates, and added a new figure panel (**fig EV3I**). We have also evaluated the thickness of the TNTs, containing or not aggregates in RD-YFP SH cells, depending on the presence of tubulin. The results are summarized in the following table.

	Tubulin-positive TNTs			Tubulin-negative TNTs		
	number counted	mean diam (um)	SD (um)	number counted	mean diam (um)	SD (um)
agg-containing TNTs	8	0,61	0,12	4	0,41	0,05
all TNTs	12	0,61	0,14	11	0,33	0,07

These results are in accordance with the work of Sartori-Rupp et al (2019) describing the ultrastructure of TNTs, and also with Onfelt et al (J Immunol, 177:8476, 2006) showing that tubulin-containing TNTs have most often a diameter around 0.6 um, whereas TNTs devoid of tubulin are thinner. We have mentioned these dimensions in the **legend of the new fig EV3I**. In the example shown in fig EV3I, we show that we could detect aggregates inside a TNT containing actin but devoid of tubulin, as well as within a thicker structure positive for both actin and tubulin. Therefore, Tau aggregates can travel through “classical” TNTs, as described before in CAD cells (Sartori-Rupp et al, 2019, Abounit S et al, Prion 10:344, 2016), as well as could be found in tubulin containing TNTs. We have now mentioned these results **in the text and discussion (page 20, lines 14-18, and page 27, lines 22-26)** of the revised manuscript. It would be interesting to further investigate the mechanisms of transfer of Tau in tubulin positive and negative TNTs, to understanding whether it is similar or different, however this will require a new full study that we believe is outside the scope of this work.

7. The work has been focused on the role cell-cell contact and little is commented about other ways for tau transfer from cell to cell. In this area, different analyses have been extraordinarily focused only in one event, and sometimes there are some controversies from work to work. For example, at the same time of reading the present manuscript, I also read Chen X. et al. 2020 Mol Neurod 6:15 or Ugboode C et al. 2019

J.Biol.Chem 294, 18967. In those works, the role of autophagy lysosomes or endosomes appear to be important, but in the present work, it is indicated that tau aggregates are essentially not associated to endosomes, lysosomes, ... This point should be further explained to indicate common or different ways for tau propagation.

We completely agree with the reviewer about the discrepancies between different works regarding propagation of Tau. Each study basically uses different cell types (often non-neuronal cells, like in the papers cited by the reviewer), which can behave differently regarding the toxic effect of aggregates as well as their ability to secrete fibrils. As discussed in point 4, we were not able to detect secretion of K18 fibrils (nor RD-YFP aggregates) in our cell culture model. Regarding the exogenous K18 fibrils, in accordance with data of the literature, including those cited by the reviewer, following internalization, K18 fibrils were found in lysosomes, also in our hands. This is consistent with an endocytic pathway that would normally end up in lysosomes. However, in this paper, we focused on addressing the localization and fate of the newly formed endogenous aggregates that in our opinion are the most relevant for modeling the real disease. Our data are consistent with such aggregates not being associated to endosomes or lysosomes, and failing degradation by autophagy. However, we did not address whether manipulating autophagy (as in Chen's paper cited by the reviewer) would affect propagation of the aggregates. Neither did we specifically address whether seeding occurred freely in the cytoplasm, in LLPS droplets, or in close proximity with vesicle membranes. These are all interesting hypothesis that we believe are outside the scope of the current paper and that will be the subject of future studies.

However, to highlight the importance of the point made by the reviewer, we have included **several paragraphs in the discussion (page 29 lines 7-12, and page 28 lines 5-18)**.

8. It was indicated that tau deposits co-labeled with ubiquitin. Thus, it will be of interest if ubiquitin is covalently bound to tau protein, or not. This analysis could be carried out by WB, and it could suggest if more than one ubiquitin molecule could be bound to tau. It was shown that tau aggregates were resistant to degradation by autophagy, but little was discussed about proteasome degradation.

We agree with the reviewer that degradation of fibrils, and their labeling by ubiquitin is of high interest. Regarding the respective degradation of the aggregates by autophagy or proteasome, we showed in figure 3 and fig EV3 the lack of effect of the treatment with bafilomycin or with bortezomide (inhibiting proteasome) on colocalization of the aggregates with p62, LAMP and LC3. We agree with the reviewer that this did not directly assess whether the fibrils could be degraded, at least partially, by proteasomes. To further study this possibility, we have quantified the number of green aggregates depending on the condition (NT, Bafilomycin or bortezomide). The results (**new fig EV3H**) clearly show that the amount of aggregates is not affected when proteasome is inhibited, therefore excluding proteasomal degradation.

Following the suggestion of this reviewer we have further investigated Tau aggregate ubiquitination and have obtained new data showing by immunofluorescence that RD-YFP aggregates (induced by K18 or AD-derived brain extract) are partially co-labeled with anti-p62 and with an antibody specifically recognizing the K63-linked chains of ubiquitin (**new fig EV3B**). This type of ubiquitin chain is not associated to proteasomal degradation, but rather targets

autophagy cargos, in perfect accordance with our results. To further show that ubiquitin directly decorates the aggregates, we separated the soluble and insoluble fractions of cell extracts by ultracentrifugation, and showed that the insoluble aggregates are also recognized by the anti-ubiquitin antibody in WB analysis (**new fig EV3C**). The results of these experiments strongly suggest that aggregates are covalently bound to ubiquitin chains. Thus, we hope that the reviewer would agree with our conclusions. We have addressed this point further in **pages 29 (lines 24-26) and 30 (lines 1-5) of the discussion**.

9. In the first paragraph of the Discussion, the previous and recent works of Braak should be quoted. In the recent work of Braak is suggested that the first tau inclusions are present at the locus coeruleus.

We thank the reviewer for the comment. We modified the introduction and discussion sections of the manuscript, adding the references to these works, as well as a more recent and interesting comment by Grinberg and Heinsen (2017) (**pages 3 and 25, lines 11-12 and 4-7 respectively**).

10. Although TNTs were first described (see ref 43) in "neuronal-like" cultured-PC-12 cells, little is known about their presence (and physiological or pathological role), in human brain and in tauopathies like Alzheimer disease. In this work, the analyses have been mainly carried out in cell culture but not in any "in vivo model".

In summary, the technical quality of the analyses done is high but the complementation of the data with "in vivo" studies will greatly improve the meaning of the work.

We would be very happy to be able to detect TNTs in human brains of patients or in mice models. Regrettably, this is still not possible in the absence of specific markers that could distinguish TNTs in such complex tissues. Plus, the fragility of these structures would require specific fixation protocols to preserve them. The best model developed so far to answer this point is the organotypic mouse brain slices that we used in fig 5. This is a model that our group has already used to show transfer of a-synuclein aggregates (Loria et al, Acta Neuropathol 134: 789, 2017). Our group has also recently brought to light TNT-like connections in primary neurons (Vargas et al, EMBO J 38: e101230, 2019), which were also used in this work. On the other hand, as suggested by reviewer 3, to improve the significance of our work, we have obtained consistent data using brain extracts of an AD patient as a source of fibrils, instead of K18 fibrils (**new fig 2F, 3D, E and fig EV3B, and movies EV4 and EV5**). See also points 1 and 11.

Referee #3 (Comments on Novelty/Model System for Author):

They have developed a nice model system. It would be important to evaluate the efficacy of the system using AD derived tau fibrils.

Referee #3 (Remarks for Author):

EMM-2020-12025 Fate and propagation of endogenously formed Tau aggregates in neuronal cells

The goal of the study by Chastagner et al is to develop a neuronal model capable of tracking the fate of an exogenously provided tau fibril- starting from getting internalized by the cell, to aggregating and seeding endogenous tau proteins, lastly followed by its spread to a neighboring cell via tunneling nanotubes (TNT). The authors also show that tau aggregates, although colocalizes with p62, do not undergo clearance via lysosomal pathway instead, they discard this unwanted aggregated protein principally through TNT, resulting in its spreading. Finally, they culminate their study by verifying their cell to cell contact theory in two coculture system involving tau fibril containing CAD cells as donors to MAP2 positive neuronal cultures and hippocampal organotrophic cultures respectively.

The following study provides novel insights into how tau fibril can aggregate, seed and transfer its toxicity to neighboring cells in a monoculture system.

Comments:

11. The most important outreach of the model proposed in this study is in its applicability towards understanding toxic tau propagation in diseases like Alzheimer's (AD). For a while it has been known that heparin induced tau aggregation do not form AD specific tau aggregates (or aggregates resembling any other tauopathy). Hence, the authors need to use a more physiologically relevant tau fibril in their model (eg- tau fibril extracted from AD brain).

We thank the reviewer for this remark, it was very helpful to increase the relevance of this study. We used brain extracts from an AD patient (already described and used in Sanders et al, Neuron 82: 1271, 2015) to check whether seeding of RD-YFP in our SH-SY5Y cells was induced as with K18 fibrils. As shown in **new fig 2F, and movies EV4 and EV5** we could visualize seeding and obtained the kinetic parameters using Incucyte analysis. Although the efficiency was lower compared to K18 (as expected), the kinetics of appearance of RD-YFP aggregates over time was similar, strongly suggesting that it followed the same mechanism. In addition, co-labeling experiments (**new fig 3D, E and fig EV3B**) show that as described previously, the RD-YFP-induced aggregates are colocalizing with p62, but not with LAMP1, suggesting that their cellular fate is the same as K18-induced aggregates. Together, these results reinforce the relevance of our model for AD studies. We have added several sentences in the discussion (**page 26, lines 4-8 and page 29, lines 24-26**) regarding these important results.

12. Since the authors used a synthetic tau fibril seed there is a possibility that other modes of transmission such as exosomes gets masked due to lack of a physiologically potent and antigenic tau fibril capable of being identified and hence loaded into an exosomal vesicle. This may explain why cells in their model preferentially uses TNT (cell to cell contact) over other non-cell to cell contact methods (this may be an artifact of their culture system caused by the choice of their toxic trigger). The authors need to re-evaluate the properties of their neuronal model system as they change their exogenous tau fibril source.

We completely agree with the reviewer that it would have been very nice to show that the AD extract-induced aggregates are propagated from cell-to-cell in a TNT-dependent manner. Although we have tried to perform coculture experiments to address this point, we were unable to obtain a donor cell population where the number of cells expressing aggregates was high enough to have good chances to transfer material to acceptor cells (differently labeled). Briefly, in our coculture experiments where aggregation of RD-YFP in donor cells was induced by K18 fibrils, we had a donor cell population where about 25% of cells expressed aggregates, making the probability to visualize transfer quite high (we calculated that 1 donor cell gave to 0,3 to 0,6 acceptor cell). In contrast, when using as donor cells RD-YFP cells treated with AD brain extracts, the number of cells expressing aggregates at the moment of the coculture was much less (less than 0,45%), and we were unable to detect transfer events over the culture (which would have represented less than $0.45 \times 0,3 = 0,135\%$ of acceptor cells, less than 30 cells in the culture in the best case). Therefore, we could not conclude on these experiments. Nevertheless, we believe that our immunofluorescence experiments showing that AD XT-induced RD-YFP aggregates have the same proportion of p62 and LAMP co-labeling as the K18-induced ones (**new fig 3E**), strongly suggesting that both types of aggregates have the same fate once inside the cell, therefore it is likely that they would also spread in a similar way. We realize that this is a weakness of this paper due to the limitations of the experiments that can be performed with the AD derived fibrils. Therefore, we have added this comment in the **discussion 7 (page 28, lines 7-11)**, citing as well other studies that have addressed this issue. We also agree with the reviewer that using FL Tau fibrils instead of K18 or RD-YFP aggregates would reinforce our model, in particular regarding the ways of propagation of Tau fibrils. We were not able to perform all these experiments for technical and time reasons. However, we should mention that previous studies from our group (Abounit et al Prion, 10: 344, 2016) and from another group (Tardivel et al, Acta Neuropathol. Commun 4: 117, 2016,) had detected FL Tau fibrils (2N4R and 1N4R respectively) in TNTs of CAD and neuronal cells, suggesting that this way of propagation between cells could be also involved in the case of FL Tau spreading. We added this important comment in the **discussion (page 27, lines 17-21)**.

22nd Jun 2020

Dear Prof. Zurzolo,

Thank you for the submission of your revised manuscript to EMBO Molecular Medicine. We have now received the enclosed report from the three referees who were asked to re-assess it. You will see from the comments below that while referee #2 and #3 are satisfied with the revision, referee #1 thinks that several important issues remain.

In principle, our editorial policy only allows a single round of major revision. However, after discussing with my colleagues in the editorial team, we all agree that it is important to address the referee's concerns with regards to the experiments and analysis performed with physiologically relevant human AD derived tau (the referee provides specific suggestions here). Since direct clinical relevance is key for publication at EMBO Mol Med, we think that addressing these concerns adequately is essential for acceptance of the manuscript.

We would therefore ask you to address this point (together with other issues raised by the referee) in an exceptional second round of revision. In case you need more time for the revision, please feel free to let us know.

I look forward to seeing a revised form of your manuscript as soon as possible.

Sincerely,
Jingyi

Jingyi Hou
Editor
EMBO Molecular Medicine

*** Instructions to submit your revised manuscript ***

** PLEASE NOTE ** As part of the EMBO Publications transparent editorial process initiative (see our Editorial at <https://www.embopress.org/doi/pdf/10.1002/emmm.201000094>), EMBO Molecular Medicine will publish online a Review Process File to accompany accepted manuscripts.

To submit your manuscript , please follow this link:

Link Not Available

When submitting your revised manuscript , please include:

- 1) a .docx formatted version of the manuscript text (including Figure legends and tables). Please make sure that the changes are highlighted to be clearly visible to referees and editors alike.
- 2) separate figure files*
- 3) supplemental information as Expanded View and/or Appendix. Please carefully check the authors guidelines for formatting Expanded view and Appendix figures and tables at <https://www.embopress.org/page/journal/17574684/authorguide#expandedview>
- 4) a letter INCLUDING the reviewers' reports and your detailed responses to their comments (as Word file)

Also, and to save some time should your paper be accepted, please read below for additional information regarding some features of our research articles:

- 5) The paper explained: EMBO Molecular Medicine articles are accompanied by a summary of the articles to emphasize the major findings in the paper and their medical implications for the non-specialist reader. Please provide a draft summary of your article highlighting
 - the medical issue you are addressing,
 - the results obtained and
 - their clinical impact.

- 6) For more information: There is space at the end of each article to list relevant web links for further consultation by our readers. Could you identify some relevant ones and provide such information as well? Some examples are patient associations, relevant databases, OMIM/proteins/genes links, author's websites, etc...

- 7) Author contributions: the contribution of every author must be detailed in a separate section (before the acknowledgments).

- 8) EMBO Molecular Medicine now requires a complete author checklist (<https://www.embopress.org/page/journal/17574684/authorguide>) to be submitted with all revised manuscripts. Please use the checklist as a guideline for the sort of information we need WITHIN the manuscript as well as in the checklist. This is particularly important for animal reporting, antibody dilutions (missing) and exact p-values and n that should be indicated instead of a range.

- 9) Every published paper now includes a 'Synopsis' to further enhance discoverability. Synopses are displayed on the journal webpage and are freely accessible to all readers. They include a short stand first (maximum of 300 characters, including space) as well as 2-5 one sentence bullet points

that summarise the paper. Please write the bullet points to summarise the key NEW findings. They should be designed to be complementary to the abstract - i.e. not repeat the same text. We encourage inclusion of key acronyms and quantitative information (maximum of 30 words / bullet point). Please use the passive voice. Please attach these in a separate file or send them by email, we will incorporate them accordingly.

You are also welcome to suggest a striking image or visual abstract to illustrate your article. If you do please provide a jpeg file 550 px-wide x 400-px high.

10) A Conflict of Interest statement should be provided in the main text

11) Please note that we now mandate that all corresponding authors list an ORCID digital identifier. This takes <90 seconds to complete. We encourage all authors to supply an ORCID identifier, which will be linked to their name for unambiguous name identification.

Currently, our records indicate that the ORCID for your account is 0000-0001-6048-6602.

Link Not Available

12) The system will prompt you to fill in your funding and payment information. This will allow Wiley to send you a quote for the article processing charge (APC) in case of acceptance. This quote takes into account any reduction or fee waivers that you may be eligible for. Authors do not need to pay any fees before their manuscript is accepted and transferred to our publisher.

Photos 400-800 DPI

*Additional important information regarding figures and illustrations can be found at <http://bit.ly/EMBOPressFigurePreparationGuideline>

***** Reviewer's comments *****

Referee #1 (Comments on Novelty/Model System for Author):

The revised manuscript raises more questions than it answers, and diminishes enthusiasm for the work. In particular, they do not show that human AD derived tau can seed from donor to recipient

cells; they do not have biochemical evidence for tau ubiquitylation (etc). The first is the key point, as pointed out in the initial review - the relevance of the work is increased if they expand beyond artificially aggregated K18 to biologically derived tau

Referee #1 (Remarks for Author):

Review of resubmission of Chastagner et al

The experiment using human brain extract failed due to the low rate at which the donor cells have aggregates. This is not an adequate answer to the important criticism that more physiologically relevant tau should be studied; indeed multiple ways of increasing the yield in the donor cells are possible (cell sort to enrich; use more potent source of seeds, increase the concentrations, use lipofectamine, etc). This would enable addressing a second unaddressed criticism- allowing an antibody blocking study in that setting.

The response to whether antibody treatment is appropriate is also inadequate. The authors argue that there isn't a good antibody to use for K18, and that the K18 isn't present in the media and so isn't secreted. But the biologically relevant form is not heparin induced K18 fibrils, it is full length tau, which is present in the extracellular medium and which can be taken up by neurons. The authors partially expand their study to use AD derived tau, which is an improvement, but fail to do the critical experiments with this reagent.

The description of the AD brain extract needs to include details on how it was prepared, the concentration of tau, and the experiments should have a negative control - from a control human brain. The conclusion that k18 is 50 fold more potent - when the molar concentration of fibril inducing tau in the extract is not known - is impossible to interpret.

The studies showing immunohistochemical co-localization with RD-YFP aggregates of, eg ubq or p62, do not show that the tau is ubiquitylated - there is every likelihood that the aggregates contain multiple proteins, any of which could be immunoreactive. Biochemical methods such as IP need to be performed. Further, the experiment asking if there is colocalization of YFP with LAMP1 is somewhat misleading - as the authors point out, the half life of the protein in a lysosome might be fairly short, and YFP can be quenched/degraded as well, so that the steady state appearance of colocalization is not helpful to tell if some RD-YFP is being degraded via autophagy. The inhibitor experiments do not give full dose response curves and are somewhat indirect. Indeed the results would be quite surprising given a large literature supporting autophagy as one mechanism of tau clearance in multiple systems.

The abstract is now misleading in that they imply that the AD extract can spread to neighboring cells etc. when this was not demonstrated.

Referee #2 (Comments on Novelty/Model System for Author):

It is very focused on a specific features of taupathology.

Referee #2 (Remarks for Author):

The authors answered properly all the points raised by this reviewer.

Referee #3 (Remarks for Author):

EMM-2020-12025-V2

This is a revised paper. The authors have satisfactorily addressed my concerns.

Thank you for the submission of your revised manuscript to EMBO Molecular Medicine. We have now received the enclosed report from the three referees who were asked to re-assess it. You will see from the comments below that while referee #2 and #3 are satisfied with the revision, referee #1 thinks that several important issues remain.

Indeed we have addressed correctly the criticisms raised by reviewers 2 and 3, since both wrote that we answered properly all the points raised. On the other hand, we are quite surprised that most of the new criticisms of reviewer 1 revolve around points raised by the two other reviewers.

In principle, our editorial policy only allows a single round of major revision. However, after discussing with my colleagues in the editorial team, we all agree that it is important to address the referee's concerns with regards to the experiments and analysis performed with physiologically relevant human AD derived tau (the referee provides specific suggestions here). Since direct clinical relevance is key for publication at EMBO Mol Med, we think that addressing these concerns adequately is essential for acceptance of the manuscript.

We thank the editor for this opportunity but we would like to remark that we have already performed the experiments requested by reviewers 2 and 3 towards providing physiologically relevant data. We note that the specific suggestions provided here by reviewer 1 towards this point (not mentioned in the first revision) are either not feasible or clearly out of the scope of this manuscript. Nonetheless, we performed additional experiments aimed to reinforce the relevance of our model and now provide new data along these lines.

In order to answer the point regarding the use of physiologically relevant human AD-derived tau (as requested by the editor and the reviewers) in our revised version of the paper we studied human AD-derived tau. We observed that endogenously formed aggregates are seeded in a comparable manner with K18 and AD fibrils, with the same localization and cell fate. Importantly, our experimental approach and results satisfied the two reviewers who originally asked for these experiments.

We have also determined that K18 fibrils seed full-length Tau, as well as colocalization of the latter, in acceptor cells, as requested by reviewer 1 (see fig EV2A-C). To extend these important findings we have performed additional experiments that are now included in the manuscript (**fig EV2D and E**) showing cell contact-dependent transfer of full-length Tau, and its localization inside TNTs. These data further confirm that full-length and RD Tau behave in a similar manner regarding both seeding and propagation in cell culture. Together these data indicate that both K18 fibrils and our cell neuronal reporter are excellent models to investigate the biology of tau aggregation and spreading of the endogenously formed aggregates, which has not been previously addressed.

Regarding the ubiquitination experiments that reviewer 1 criticizes, they were not requested by him/her, but instead by reviewer 2, who now considers this new data satisfactory. We hope to avoid "double jeopardy" here, and further emphasize that the additional requests from referee 1 are not feasible (see also below for a detailed answer).

Finally, the main point raised by Reviewer 1 was the use of an antibody in coculture experiments to see whether this would affect transfer. We do not consider this request appropriate as we provided evidence that we do not detect any fibrils in the medium. Nonetheless since the reviewer was not satisfied by our findings, we have now performed an experiment monitoring the cell-contact-dependent transfer of FL Tau in the presence of anti-Tau antibodies. We did not detect any effect of the treatments on FL Tau transfer, consistent with our demonstration that transfer occurs mainly directly from cell to cell. We added this experiment here as an **additional figure**, although we don't feel it improves the manuscript, or is within the main scope of this work.

We would therefore ask you to address this point (together with other issues raised by the referee) in an exceptional second round of revision. In case you need more time for the revision, please feel free to let us know.

We have replied in detail to the new criticisms from referee 1 below and made changes and addition in the manuscript that we hope will satisfy you and will make our work suitable for EMM.

***** Reviewer's comments *****

Referee #1 (Comments on Novelty/Model System for Author):

The revised manuscript raises more questions than it answers, and diminishes enthusiasm for the work. In particular, they do not show that human AD derived tau can seed from donor to recipient cells; they do not have biochemical evidence for tau ubiquitylation (etc). The first is the key point, as pointed out in the initial review - the relevance of the work is increased if they expand beyond artificially aggregated K18 to biologically derived tau.

In the experiments presented in the revised version of the manuscript we now use a physiological source of Tau fibrils (as recommended by this and other reviewers) to induce seeding in our cell culture model. We have also shown that the endogenously formed aggregates are seeded in a comparable manner by K18 and AD fibrils, and have the same localization and fate in the cells.

Specifically, we have demonstrated that AD patient-derived Tau seeds new fibrils in the reporter cell model (fig 2F, 3D, 3E) as acknowledged by reviewers 2 and 3.

Ubiquitylation experiments (criticized by reviewer 1) were requested by reviewer 2, who considered our new data satisfactory. It doesn't seem reasonable now to be penalized by reviewer 1 for these same experiments.

We have shown that RD-YFP aggregates, resulting from K18 or AD seeding, are co-labeled by ubiquitin, and in particular K63-linked chains. We have also provided biochemical data showing that insoluble material, positive for RD-YFP, is also ubiquitylated. Together, these results are consistent with Tau aggregates, or factors belonging to the same insoluble complexes, being ubiquitylated and becoming targets for autophagy. Since the complexes

would be engulfed by the autophagy machinery, it is unclear how the presence or absence of detectable post-translational modifications specifically on Tau aggregates would change our fundamental conclusions. Nevertheless, we have modified the text of the discussion to reflect these possibilities.

Please see also below for specific answers.

Referee #1 (Remarks for Author):

Review of resubmission of Chastagner et al

The experiment using human brain extract failed due to the low rate at which the donor cells have aggregates. This is not an adequate answer to the important criticism that more physiologically relevant tau should be studied; indeed multiple ways of increasing the yield in the donor cells are possible (cell sort to enrich; use more potent source of seeds, increase the concentrations, use lipofectamine, etc). This would enable addressing a second unaddressed criticism- allowing an antibody blocking study in that setting.

We agree with the reviewer that to be performed, these experiments need to use a higher percentage of donor cells containing endogenously formed aggregates. However, the solutions proposed by the reviewer will not help for several reasons:

1. The cell sorting proposed by this reviewer unfortunately does not allow discrimination between cells expressing soluble RD-YFP (which are green) from cells with RD-YFP aggregates (same fluorescence);
2. Regarding the source of fibrils, the extract that we used and that was fully characterized in Sanders et al, 2014, was among the most efficient tested in the laboratory. We have already tested several concentrations of extracts and chosen the most efficient for seeding.
3. Finally, we have used Lipofectamine to facilitate cell entry of AD brain material in all our experiments, since we have observed that K18 fibrils do not enter the cells efficiently to seed without Lipofectamine (see fig EV2A and B). We made the same observation for AD extracts, and tested several quantities of Lipofectamine, extracts and protein:Lipofectamine ratios.

The response to whether antibody treatment is appropriate is also inadequate. The authors argue that there isn't a good antibody to use for K18, and that the K18 isn't present in the media and so isn't secreted. But the biologically relevant form is not heparin induced K18 fibrils, it is full length tau, which is present in the extracellular medium and which can be taken up by neurons. The authors partially expand their study to use AD derived tau, which is an improvement, but fail to do the critical experiments with this reagent.

The reviewer states that fibrils formed by full length Tau are found outside cells, but to our knowledge this has not been previously demonstrated clearly to be a common phenomenon, whereas soluble Tau has been reported by many investigators. In our hands too, FL Tau previously transfected in CAD cells, was detected in a small amount in cell supernatant (see **accompanying figure, panel A**).

Furthermore, we would like to stress that in our cell model, we have not observed any seeding by treating the cells with the AD-derived extract in the absence of Lipofectamine,

therefore it is probable that the FL Tau fibrils present in this extract enter the cells through the same mechanism, Lipofectamine-induced membrane permeabilization, as K18 fibrils. As demonstrated in fig EV4A, no K18 fibrils are detectable in the cell culture medium. Even if FL Tau could be secreted in the culture medium, it most probably would not be taken up by our cells, since coculture experiments are performed in the absence of Lipofectamine, after the cells have been trypsinized and washed. This was confirmed by the new experiments presented in **fig EV2E and accompanying figure, panel A**, showing that no transfer was detected when applying cell supernatant to acceptor cells, although some FL Tau was detected by WB in the supernatant.

Regarding the better relevance of full-length Tau compared to RD Tau, in the revised version we have also determined that K18 fibrils seed and colocalize with FL-tau in the acceptor cells as requested by reviewer 1 (fig EV2A, B, C). In addition to these experiments we have now added in the new **fig EV2D, E**, quantitative data showing that FL Tau aggregates, endogenously formed under K18 treatment, mainly transfer between cells in a cell contact-dependent manner, and can be observed in TNTs. These new experiments demonstrate that FL Tau aggregates behave in the exact same manner as the RD-YFP aggregates of our cell model.

As requested by the reviewer, we have also performed the same coculture experiment in the presence of various antibodies recognizing FL-Tau-YFP chimera in different domains: Tau5, S262 (see **panel A of accompanying figure**), GFP, as well as a control antibody (TfR). Our results (**panel B of accompanying figure here below**) show that the presence of these antibodies in the cell medium did not affect transfer of FL Tau. Interestingly, we detected some FL Tau in the coculture medium, whether the cells were treated with K18 fibrils or not, using the same antibodies (**panel A of accompanying figure**). This result is in accordance with the literature showing that FL Tau aggregates can be secreted, although we cannot exclude that these molecules derive from dead cells. Either way, this result shows that antibodies were unable to inhibit transfer, even if reactive species were present in cell supernatants. This therefore strengthens our conclusions that transfer of FL Tau mainly occurs through direct cell contact, not accessible to antibodies, as it is the case for transfer of K18 or RD-YFP aggregates.

The description of the AD brain extract needs to include details on how it was prepared, the concentration of tau, and the experiments should have a negative control - from a control human brain. The conclusion that k18 is 50 fold more potent - when the molar concentration of fibril inducing tau in the extract is not known - is impossible to interpret.

As previously mentioned here and in the manuscript, the AD brain extract that we have used was already described in detail in Sanders et al (2014). It is true that we have not used a control brain extract from a "healthy" subject, but instead used non-treated cells as control, where no spontaneous aggregation was observed in the course of the experiment. Given the high specificity of the biosensors cells, if with such biological material we had observed some induced aggregation, the conclusion would be that the individual in question was probably pre-symptomatic, with fibrils at undetectable levels.

In the precious biological material derived from AD patient, the concentration of Tau was not assessed, since it would not give any information on the actual concentration of fibrils, which could be of various lengths and conformations. It is the same for the K18 fibrils, the concentration of which refers to the equivalent monomer concentration and not to the

actual fibril concentration. The quantification performed in the experiment the reviewer refers to is based on the outcome of the treatment (the number of RD-YFP aggregate-containing cells). We have not written that **K18 is 50-fold more potent**, we wrote in the results section: “Overall, ~50-fold more cells contained aggregates after three days when treated with K18 compared to AD extract”, which does not imply that one type of fibrils is more efficient than the other, precisely because we cannot compare their respective concentration. However, we realized that the way we refer to these results in the discussion (“Although seeding with the synthetic fibrils was more efficient,”) was indeed misleading and needed to be adjusted. We changed it to “Although the number of seeded cells was higher with the synthetic fibrils “. We thank the reviewer for the comment.

The studies showing immunohistochemical co-localization with RD-YFP aggregates of, eg ubq or p62, do not show that the tau is ubiquitinated - there is every likelihood that the aggregates contain multiple proteins, any of which could be immunoreactive. Biochemical methods such as IP need to be performed. Further, the experiment asking if there is colocalization of YFP with LAMP1 is somewhat misleading - as the authors point out, the half life of the protein in a lysosome might be fairly short, and YFP can be quenched/degraded as well, so that the steady state appearance of colocalization is not helpful to tell if some RD-YFP is being degraded via autophagy. The inhibitor experiments do not give full dose response curves and are somewhat indirect. Indeed the results would be quite surprising given a large literature supporting autophagy as one mechanism of tau clearance in multiple systems.

The experiments performed regarding ubiquitinylation of Tau aggregates were aimed to address concerns of Reviewer 2 (who was satisfied with our experiments). We agree with the reviewer that our demonstration is not perfect, but we believe that the experiment suggested by this reviewer is not feasible for reasons detailed below. Indeed, we would be very happy if the reviewer provided us with the detailed protocol allowing to show that only the aggregates are ubiquitinated.

Before performing an IP to purify YFP-reactive molecules, it would be necessary to separate insoluble from soluble material (both composed of RD-YFP molecules), otherwise the only demonstration would be that RD-YFP material (soluble and/or insoluble) is ubiquitinated. To separate soluble from insoluble material, an ultracentrifugation step is necessary (as we performed). Once done, the insoluble material would need to be re-solubilized before the immunoprecipitation step. In a regular situation, this could be done by using mild detergent solubilization, followed by dilution in order not to impair antigen-antibody recognition nor to denature antibodies. In our attempts to solubilize RD-YFP fibrils, and in accordance with the literature on Tau fibrils, even 4% SDS is not enough to completely dissociate the fibrils, and we had to additionally treat with urea. In these conditions it is impossible to perform immunoprecipitation, even after substantial dilution.

Thus, we agree with the reviewer that it is possible that associated proteins inside the aggregates could be the targets of ubiquitination and responsible for immunoreactivity. However, if they are part of the same aggregate we can reasonably assume that this signal will induce the fate of the entire aggregate. We have changed the text to be more accurate, since our results indeed show that the aggregates themselves, or proteins belonging to the same insoluble complexes, are ubiquitinated. To our knowledge, our demonstration corresponds to what is generally provided in the literature.

We do not understand the point regarding LAMP1, and whether the reviewer considers that using classical and recommended lysosomal inhibitors is not useful to demonstrate whether proteins can reach lysosomes for degradation.

Regarding the “large literature supporting autophagy as one mechanism of tau clearance in multiple systems” claimed by the reviewer, to our knowledge it refers mostly to exogenous fibrils, as discussed in the discussion section, and not to endogenously formed aggregates.

The abstract is now misleading in that they imply that the AD extract can spread to neighboring cells etc. when this was not demonstrated.

We agree with the reviewer and we have substantially modified the abstract as well as the end of introduction to be more specific about our data.

In conclusion, we would like to point out that this paper is about describing a very useful and relevant cell model to study how tauopathies develop and spread. Specifically addressing the intracellular fate and spreading of endogenously formed fibrils, which has not been addressed before. In the revised version of the manuscript we have added significant results, addressing all reviewers’ criticisms concerning the in vivo relevance of our model. We have now performed new experiments, which strongly suggested that FL Tau formed aggregates and propagated similarly to RD-YFP, reinforcing the relevance of our model and of findings.

Referee #2 (Comments on Novelty/Model System for Author):

It is very focused on a specific features of taupathology.

Referee #2 (Remarks for Author):

The authors answered properly all the points raised by this reviewer.

Referee #3 (Remarks for Author):

EMM-2020-12025-V2

This is a revised paper. The authors have satisfactorily addressed my concerns.

Accompanying Figure for reviewer

A: Tau5 and S262 antibodies recognize FLTau1N4R P301S-YFP expressed in CAD cells

B: Tau5, S262 and GFP antibodies do not affect FLTau1N4R P301S-YFP cell contact-dependent transfer

Culture condition of acceptor cells with donor cells	cocult	SN	cocult	cocult	cocult	cocult
Antibody	-	-	Tau5	S262	GFP	TfR

Legend to accompanying figure

A. Tau5 and S262 antibodies recognize FLTau1N4R P301S-YFP expressed in CAD cells.

Western blots using successively Tau-5 antibody (mouse monoclonal, aa210-241, outside RD domain), left panel, and S262 antibody (rabbit polyclonal against a peptide surrounding S262, in RD domain), right panel. At the end of a 24-hour coculture (donor cells: FLTau1N4R P301S-YFP transfected cells, treated or not with K18, plus acceptor H2B-mcherry cells), supernatants were collected, centrifuged at low speed to remove cells, and next ultracentrifuged at 100000g. Pellets were solubilized in 8M urea and loaded on 4-12% gel in MES buffer. In parallel cell extracts of the FLTau1N4R P301S-YFP transfected cells were prepared and loaded on the same gel. Note that the loaded materials from SN and extracts correspond to comparable number of donor cells. For each WB, white lane between SN and XT corresponds to intercalating lanes on the same gel that have been spliced out. Below the blots are indicated the exposure time necessary to get the signal of the figure. These WB indicate that both antibodies are able to detect FLTau1N4R P301S-YFP in cell extracts. In addition, few quantities of FLTau1N4R P301S-YFP were also detected in cell supernatant with the more sensitive antibody (S262).

B: Tau5, S262 and GFP antibodies do not affect FLTau1N4R P301S-YFP cell contact-dependent transfer

Below the schematic representation of the experiment, which was performed as described in new fig EV2D, is the quantification of the percentage of acceptor cells containing green dots. Antibodies were added when coculture was started for 24 hours, at 2 ug/ml. TfR antibody was used as rabbit control antibody. GFP antibody was also made in rabbit (Invitrogen). SN indicates that acceptor cells received supernatant from K18-treated donor cells (no green dot detected). The number of acceptor cells analyzed in each condition was 164, 158, 64, 72, 117, 92 respectively.

24th Aug 2020

Dear Chiara,

Thank you for the submission of your revised manuscript to EMBO Molecular Medicine. We have now received the enclosed report from the referee who was asked to re-assess it. As you will see the referee is now supportive and I am pleased to inform you that we will be able to accept your manuscript pending the following amendments:

1. Please discuss the potential limitations of the model system according to the referee's comment.
2. In the main manuscript file, please do the following:
 - Reduce the number of keywords to 5
 - remove the blue color font
 - Add callouts for Fig. 3 E & F
 - Re-order the sections of the manuscript text according to the author guidelines: move Materials & Methods after Discussion, move Author contribution after Acknowledgement, add section heading for Conflict of interest, add The Paper Explained to the Manuscript before the References, and For More information after The Paper explained.
 - in legends, provide exact $n=$ and exact $p=$ values, not a range, along with the statistical test used. Some authors find that in order to keep the figures clear, providing an appendix supplemental table with all exact p -values is preferable. You are welcome to do this if you want to.
3. Figures: please improve figure quality/resolution throughout, is available. Add scale bars to magnifications in Fig 2B, 3A, 4C, EV3B, EV5A.
4. Movies: please remove Expanded view movie legends from the manuscript file and zip together with the corresponding Movie file in Doc or txt format.
5. Data availability: since there is no large-scale dataset generated in this study, please include the following single sentence in this section- "This study includes no data deposited in external repositories".
6. Our data editor has made a couple of suggestions on your manuscript (see attached), please address.
7. As part of the EMBO Publications transparent editorial process initiative (see our Editorial at <http://embomolmed.embopress.org/content/2/9/329>), EMBO Molecular Medicine will publish online a Review Process File (RPF) to accompany accepted manuscripts.
 - a. In the event of acceptance, this file will be published in conjunction with your paper and will include the anonymous referee reports, your point-by-point response and all pertinent correspondence relating to the manuscript. Let us know if you do not agree with this.
 - b. Please note that the Authors checklist will be published at the end of the RPF.

I look forward to seeing a revised version of your manuscript as soon as possible.

Sincerely,

Jingyi

Yours sincerely,

Jingyi Hou

Jingyi Hou
Editor
EMBO Molecular Medicine

*** Instructions to submit your revised manuscript ***

To submit your manuscript, please follow this link:

Link Not Available

- 1) a .docx formatted version of the manuscript text (including Figure legends and tables)
- 2) Separate figure files*
- 3) supplemental information as Expanded View and/or Appendix. Please carefully check the authors guidelines for formatting Expanded view and Appendix figures and tables at <https://www.embopress.org/page/journal/17574684/authorguide#expandedview>
- 4) a letter INCLUDING the reviewer's reports and your detailed responses to their comments (as Word file).
- 5) The paper explained: EMBO Molecular Medicine articles are accompanied by a summary of the articles to emphasize the major findings in the paper and their medical implications for the non-specialist reader. Please provide a draft summary of your article highlighting
- the medical issue you are addressing,

- the results obtained and
- their clinical impact.

6) For more information: There is space at the end of each article to list relevant web links for further consultation by our readers. Could you identify some relevant ones and provide such information as well? Some examples are patient associations, relevant databases, OMIM/proteins/genes links, author's websites, etc...

7) Author contributions: the contribution of every author must be detailed in a separate section.

8) EMBO Molecular Medicine now requires a complete author checklist (<https://www.embopress.org/page/journal/17574684/authorguide>) to be submitted with all revised manuscripts. Please use the checklist as guideline for the sort of information we need WITHIN the manuscript. The checklist should only be filled with page numbers where the information can be found. This is particularly important for animal reporting, antibody dilutions (missing) and exact values and n that should be indicated instead of a range.

9) Every published paper now includes a 'Synopsis' to further enhance discoverability. Synopses are displayed on the journal webpage and are freely accessible to all readers. They include a short stand first (maximum of 300 characters, including space) as well as 2-5 one sentence bullet points that summarise the paper. Please write the bullet points to summarise the key NEW findings. They should be designed to be complementary to the abstract - i.e. not repeat the same text. We encourage inclusion of key acronyms and quantitative information (maximum of 30 words / bullet point). Please use the passive voice. Please attach these in a separate file or send them by email, we will incorporate them accordingly.

You are also welcome to suggest a striking image or visual abstract to illustrate your article. If you do please provide a jpeg file 550 px-wide x 400-px high.

10) A Conflict of Interest statement should be provided in the main text

11) Please note that we now mandate that all corresponding authors list an ORCID digital identifier. This takes <90 seconds to complete. We encourage all authors to supply an ORCID identifier, which will be linked to their name for unambiguous name identification.

Currently, our records indicate that the ORCID for your account is 0000-0001-6048-6602.

Please click the link below to modify this ORCID:
Link Not Available

12) The system will prompt you to fill in your funding and payment information. This will allow Wiley to send you a quote for the article processing charge (APC) in case of acceptance. This quote takes into account any reduction or fee waivers that you may be eligible for. Authors do not need to pay any fees before their manuscript is accepted and transferred to our publisher.

Photos 400-800 DPI

*Additional important information regarding figures and illustrations can be found at <http://bit.ly/EMBOPressFigurePreparationGuideline>

The system will prompt you to fill in your funding and payment information. This will allow Wiley to send you a quote for the article processing charge (APC) in case of acceptance. This quote takes into account any reduction or fee waivers that you may be eligible for. Authors do not need to pay any fees before their manuscript is accepted and transferred to our publisher.

***** Reviewer's comments *****

Referee #1 (Comments on Novelty/Model System for Author):

As noted in the review, the authors have not been able to use human AD derived tau. It may be that their model system, using recombinant fibrils and overexpressed FL tau, is sufficient to provide insight, but it is unclear at this stage if that is the case. Nonetheless, many papers have been written with similar methods, and if the authors are limited in what they can accomplish it may be that a clear paragraph in the discussion would suffice.

Referee #1 (Remarks for Author):

The authors have added an experiment that includes full length tau as part of the aggregation process, then tracks it into the next cells. This is an improvement from previous experiments, in that it uses full length tau, but the revised manuscript does not address the major concern, which is that human AD derived tau is not used for the critical experiments. The issue is that the human derived tau has different biochemical and biophysical properties in terms of uptake/seeding compared to recombinant species, as demonstrated perhaps most definitively by Dr Virginia Lee's group in injections of various tau preparations into the extracellular space in mice.

The authors assert that their conclusions about ubiquitin need not be demonstrated with more rigorous experiments, because of the difficulty in solubilizing fibrils. While the biochemical characterization of tau in fibrils, from IP to Western blots to mass spec, are used routinely in the field and have been for years, perhaps this limitation with the small amount of material available is understandable. Regardless, this is not a major point of the paper.

Despite these critiques, one has to take the authors rebuttal comments seriously. The paper is limited by the system in which they are working, and the available reagents. However, it is also clear that work with recombinant and similar systems can have a substantial amount of value and interest. Perhaps a very clear paragraph in the discussion, discussing the limitations of the model and citing the literature showing similarities and different properties of the tau they have studied

compared to AD-derived tau would suffice to have both the strengths and the weaknesses of the model system be clear to the reader.

***** Reviewer's comments *****

Referee #1 (Comments on Novelty/Model System for Author):

As noted in the review, the authors have not been able to use human AD derived tau. It may be that their model system, using recombinant fibrils and overexpressed FL tau, is sufficient to provide insight, but it is unclear at this stage if that is the case. Nonetheless, many papers have been written with similar methods, and if the authors are limited in what they can accomplish it may be that a clear paragraph in the discussion would suffice.

We agree with the reviewer and have added a paragraph at the end of the first paragraph of the discussion clearly stating the limitations of our work regarding human AD-derived fibrils (p.19, lines 8-19).

Referee #1 (Remarks for Author):

The authors have added an experiment that includes full length tau as part of the aggregation process, then tracks it into the next cells. This is an improvement from previous experiments, in that it uses full length tau, but the revised manuscript does not address the major concern, which is that human AD derived tau is not used for the critical experiments. The issue is that the human derived tau has different biochemical and biophysical properties in terms of uptake/seeding compared to recombinant species, as demonstrated perhaps most definitively by Dr Virginia Lee's group in injections of various tau preparations into the extracellular space in mice.

The authors assert that their conclusions about ubiquitin need not be demonstrated with more rigorous experiments, because of the difficulty in solubilizing fibrils. While the biochemical characterization of tau in fibrils, from IP to Western blots to mass spec, are used routinely in the field and have been for years, perhaps this limitation with the small amount of material available is understandable. Regardless, this is not a major point of the paper.

We agree with the reviewer. Furthermore, it is quite difficult to use mass spec for detecting ubiquitination events, because trypsin cuts out the ubiquitin chains at Arg74 leaving only Gly75 and Gly76 residues on the modified lysine of the substrate peptide, and only a modified workflow to enrich for and detect diGly peptides (the last two aa of ubiquitin, covalently bound to the ubiquitylated substrate) using immunopurification has been recently successfully used to uncover ubiquitinomes. However, because of the IP step, this approach would be difficult in the case of insoluble tau fibrils, and would not allow identifying the type of chains formed.

Despite these critiques, one has to take the authors rebuttal comments seriously. The paper is limited by the system in which they are working, and the available reagents. However, it is also clear that work with recombinant and similar systems can have a substantial amount of value and interest. Perhaps a very clear paragraph in the discussion, discussing the limitations of the model and citing the literature showing similarities and different properties of the tau they have studied compared to AD-derived tau would suffice to have both the strengths and the weaknesses of the model system be clear to the reader.

We added in the first paragraph of the discussion the limitations of our data and put them into perspective with the works comparing the abilities of natural and synthetic fibrils to induce Tau pathology, citing in particular V. Lee's papers, as suggested by the reviewer. We believe that this paragraph actually reinforces the interest of our model and of our results, and we thank the reviewer for his/her suggestion.

21st Sep 2020

Dear Prof. Zurzolo,

We are pleased to inform you that your manuscript is accepted for publication and is now being sent to our publisher to be included in the next available issue of EMBO Molecular Medicine.

We would like to remind you that as part of the EMBO Publications transparent editorial process initiative, EMBO Molecular Medicine will publish a Review Process File online to accompany accepted manuscripts. If you do NOT want the file to be published or would like to exclude figures, please immediately inform the editorial office via e-mail.

Please read below for additional IMPORTANT information regarding your article, its publication and the production process.

Congratulations on your interesting work,

Jingyi Hou

Jingyi Hou
Editor
EMBO Molecular Medicine

Follow us on Twitter @EmboMolMed
Sign up for eTOCs at embopress.org/alertsfeeds

***** Reviewer's comments *****

*** ** IMPORTANT INFORMATION *** **

SPEED OF PUBLICATION

The journal aims for rapid publication of papers, using the advance online publication "Early View" to expedite the process: A properly copy-edited and formatted version will be published as "Early View" after the proofs have been corrected. Please help the Editors and publisher avoid delays by providing e-mail address(es), telephone and fax numbers at which author(s) can be contacted.

Should you be planning a Press Release on your article, please get in contact with embomolmed@wiley.com as early as possible, in order to coordinate publication and release dates.

LICENSE AND PAYMENT:

All articles published in EMBO Molecular Medicine are fully open access: immediately and freely

available to read, download and share.

EMBO Molecular Medicine charges an article processing charge (APC) to cover the publication costs. You, as the corresponding author for this manuscript, should have already received a quote with the article processing fee separately. Please let us know in case this quote has not been received.

Once your article is at Wiley for editorial production you will receive an email from Wiley's Author Services system, which will ask you to log in and will present you with the publication license form for completion. Within the same system the publication fee can be paid by credit card, an invoice, pro forma invoice or purchase order can be requested.

Payment of the publication charge and the signed Open Access Agreement form must be received before the article can be published online.

PROOFS

You will receive the proofs by e-mail approximately 2 weeks after all relevant files have been sent to our Production Office. Please return them within 48 hours and if there should be any problems, please contact the production office at embopressproduction@wiley.com.

Please inform us if there is likely to be any difficulty in reaching you at the above address at that time. Failure to meet our deadlines may result in a delay of publication.

All further communications concerning your paper proofs should quote reference number EMM-2020-12025-V4 and be directed to the production office at embopressproduction@wiley.com.

Thank you,

Jingyi Hou
Editor
EMBO Molecular Medicine

Corresponding Author Name: Chiara Zurzolo
Journal Submitted to: EMBO Molecular Medicine
Manuscript Number: EMM-2020-12025